# Tracking the Impact of Urban Air Masses on Convective Precipitation: A Multi-Member Modeling Study

Friederike Keil<sup>1</sup>, Markus Quante<sup>1</sup>, Bernd Heinold<sup>2</sup>, and Volker Matthias<sup>1</sup>

<sup>1</sup>Helmholtz-Zentrum Hereon, Geesthacht, Germany

**Correspondence:** Friederike Keil (friederike.keil@hereon.de)

Abstract. Urban emissions impact aerosol-cloud interactions and thereby modify precipitation patterns, yet their absolute effects remain uncertain due to internal atmospheric variability and limitations of conventional analysis methods. This study aims to further quantify the influence of urban aerosol fields on convective precipitation through explicit chemistry-cloud coupling. Using the coupled COSMO-DCEP-MUSCAT modeling system, we simulate two convective events passing the city of Leipzig, Germany, with five-member ensemble experiments comparing total emissions to zero urban emission scenarios. Cloud droplet activation is calculated from prognostic three-dimensional aerosol fields, providing a physically consistent representation of aerosol-cloud interactions. We use backward trajectory analysis to directly trace urban air masses from convective clouds back to the region of urban emission sources, enabling objective sampling of individual clouds and isolation of local emission effects. The results reveal case-dependent responses. Under moderate atmospheric instability, urban aerosols locally modify the cloud microphysics and precipitation without altering the overall structure of the convective system. Under stronger initial instability, the urban emissions intensify the precipitation, leading to stronger downdrafts, causing a premature system decay and shorter lifetime compared to the zero urban emission scenario. Ensemble analysis demonstrates that emission-induced changes are comparable in amount to internal variability, highlighting the need for multiple realizations and significance testing. The results of this study reveal that urban aerosol effects are highly case-dependent, challenging assumptions about uniform impacts. The same air pollution source can either delay, enhance or suppress convection depending on prevailing atmospheric conditions.

# 1 Introduction

Urban areas can play a significant role in modifying precipitation patterns. More than half of the cities worldwide, along with their downwind regions, experience higher levels of precipitation compared to nearby rural areas (Sui et al., 2024). The combined effects of climate change and increasing urbanization are expected to further intensify precipitation events. As atmospheric temperatures rise, the air's capacity to hold moisture increases (Seneviratne et al., 2021). When coupled with urban expansion, this may result in more extreme short-duration rainfall events in and around large cities (Lehmann et al., 2015). The urban heat island effect and alterations in vertical wind due to urban morphology are especially important in influencing urban rainfall and have received much attention in the literature (Liu and Niyogi, 2019; Yue et al., 2021).

<sup>&</sup>lt;sup>2</sup>Leibniz Institute for Tropospheric Research, Leipzig, Germany

Moreover, urban areas are a considerable source of aerosols, primarily from traffic, residential heating, and industrial activities. Depending on their size and chemical composition, these urban aerosols can be activated as cloud condensation nuclei (CCN) or ice-nucleating particles (INP). While sulfate aerosols are effective CCN (Pruppacher and Klett, 1997), black carbon, mineral dust, and certain organic particles are more relevant as INPs (Burrows et al., 2022). Once activated, they modify the microphysical structure of clouds, which in turn affects the generation and intensity of precipitation. The overall impact of urban aerosols on rainfall is complex, as they can suppress, delay, or enhance precipitation depending on the cloud development stage and whether warm or ice-phase processes dominate.

A higher concentration of aerosols may lead to the activation of more cloud droplets (Twomey, 1974). However, the resulting cloud droplet size spectrum tends to be narrower, which reduces coalescence and collision efficiency, thereby decreasing raindrop growth. This ultimately suppresses precipitation and enhances cloud lifetime (Albrecht, 1989). This effect is particularly relevant in warm-phase clouds, where both models and observations indicate that urban aerosols have a pronounced suppressing effect on precipitation (Van Den Heever and Cotton, 2007; Rosenfeld, 2000). The initial precipitation suppression delays downdraft development, allowing more liquid cloud water to ascend to higher altitudes where temperatures are below freezing. As a result, more latent heat is released when droplets freeze, and when they fall and melt at lower levels, this heat is reabsorbed. This process intensifies ice-phase dynamics, enhances cloud buoyancy and invigorates convection, ultimately increasing precipitation amounts (Rosenfeld et al., 2008).

Especially after the publication of Rosenfeld et al. (2008), many observational and modeling studies were conducted concerning aerosol invigoration in deep convection. While warm-phase invigoration was in general supported, intensities of cold-phase invigoration differ considerably among the studies and from Rosenfeld et al. (2008). E.g. Igel and van den Heever (2021) report that for warm-based storms the updrafts are weakened instead of strengthened. In contrast, Marinescu et al. (2021) present results from a multi-model comparison investigating the impact of varying CCN concentrations on updrafts in deep convection. All models simulate stronger mean updrafts at altitudes between 4 and 7 km under high CCN conditions. The type of models evaluated is close to that used in the present study. A critical assessment of the state of aerosol invigoration topic is provided in an opinion paper by Varble et al. (2023).

Building on the concept of aerosol invigoration, several studies have examined how urban aerosols influence precipitation patterns. Model-based studies indicate that the enhancement of precipitation tends to occur downwind of the city (Han et al., 2012; Kawecki et al., 2016; Sarangi et al., 2018). Consequently, the aerosol invigoration effect leads to a decrease in early-stage rain and an intensification of heavy rainfall, which was found in model and observational studies (Schmid and Niyogi, 2017; Zheng et al., 2020).

Apart from influencing microphysics, light-absorbing aerosols, such as soot or dust, directly influence radiative effects by backscattering or absorbing sunlight. These radiative effects tend to cool the surface by reducing net short-wave radiation at the surface and heating the lower atmosphere through absorption of radiation. This can result in a stabilization of the lower atmosphere, which weakens the updraft and reduces precipitation rates (Fan et al., 2015; Talukdar et al., 2019).

Multiple single-event-based analyses have proven to be useful, offering a detailed examination of urban aerosol impacts under real-world conditions. Kawecki et al. (2016) studied a meso-scale convective system passing Kansas City, Missouri. They

found that aerosol concentration levels significantly influenced the development of the convective system and shifted the precipitation patterns. Similarly, Sarangi et al. (2018) demonstrated that the urban area of Kanpur, India, and downwind regions experience increased precipitation rates and that the aerosol loading positively correlates with this increase. Further research by Fan et al. (2020) focused on a convective storm near Houston, Texas, highlighting that the urban aerosol effect is more significant than the effects through the changes in land cover types for coastal cities. While the urban land cover effect accelerates rain initiation and slows down the storms dissipation, the convective intensity and precipitation are increased by urban aerosols through aerosol-cloud interactions. Van Den Heever and Cotton (2007) investigated the development of a convective storm and precipitation over and downwind of St. Louis, Missouri. Their study demonstrated that urban-enhanced aerosol concentrations significantly affect convective storm dynamics and precipitation development in downwind areas. The study highlights that the influence of urban aerosols is more pronounced in less industrialized regions, where lower background aerosol concentrations amplify their impact on convection and precipitation processes.

Although focusing on a single event limits the generalizability of their findings to other meteorological contexts or regions, such process-level studies are essential for improving our understanding of aerosol-cloud interactions. Collectively, these studies highlight the significant and complex role that urban environments and aerosols play in shaping local weather patterns and influencing precipitation dynamics.

Moreover, these studies have examined individual events under specific meteorological conditions only using a single simulation (Sarangi et al., 2018; Fan et al., 2020; Van Den Heever and Cotton, 2007) or three simulations (Kawecki et al., 2016) per emission scenario. However, Varble et al. (2023) highlighted the necessity of using ensembles with perturbed initial or boundary conditions rather than relying on a single simulation to better capture the sensitivity and robustness of model results. Another significant challenge is isolating the urban aerosol-cloud effect from other factors, such as the roughness and thermal effects of urban morphology (Rozoff et al., 2003). Quantifying the specific impact of urban aerosols on clouds and precipitation is particularly difficult due to the relatively small signal amidst high natural variability.

To address these challenges, we investigate how urban emissions can modify precipitating cloud systems while accounting for the natural meteorological variability. We perform highly resolved ensemble simulations of two convective events using the online-coupled model system COSMO-MUSCAT-DCEP (Wolke et al., 2004, 2012), integrating urban parameterization and direct aerosol-cloud coupling. The model employs a two-moment microphysics scheme, which predicts both mass and number concentration of hydrometeors. Specifically, to account for the influence of aerosols on cloud droplet formation, we integrated MUSCAT-simulated aerosol fields into the microphysics scheme.

This study focuses on convective events over eastern Germany. The region is characterized by a mix of urban and industrial emission sources and an overall relatively high background aerosol concentration. We specifically selected convective events that pass through the urban emissions plume of Leipzig, a medium-sized town in the eastern German state of Saxony, providing ideal case studies to investigate the influence of urban aerosols on cloud development and precipitation formation, allowing for a detailed process-level analysis under realistic meteorological conditions.

We trace the transport pathways of polluted urban air masses from their emission sources into regions of cloud and precipitation formation by analyzing Lagrangian back trajectories. In addition, we use an ensemble modeling framework to capture the

95 range of possible atmospheric responses and to enhance the robustness of our findings.

By analyzing these events, we aim to deepen our understanding of the complex interactions between urban emissions, cloud microphysics, and precipitation development. This knowledge is crucial for addressing urban-induced alterations of heavy precipitation events.

#### 2 Methods

#### 100 **2.1 Model**

We use version 5.05 of the non-hydrostatic regional numerical weather prediction model COSMO (COnsortium for Small scale MOdelling) (Schättler et al., 2018), online-coupled to the chemistry transport model MUSCAT (MUltiScale Chemistry Aerosol Transport) (Wolke et al., 2004, 2012). MUSCAT, driven online by the meteorological model COSMO, simulates the transport and chemical transformation of gas-phase species and aerosol populations using a system of three-dimensional, timedependent advection-diffusion-reaction equations. Aerosol dynamics are represented by a hybrid bulk-bin scheme, which is the standard configuration for air quality simulations. This scheme includes 25 prognostic particle tracers: primary PM2.5 and PM10, primary organic carbon (POC), elemental carbon (EC), sulfate, nitrate, ammonium, secondary organic aerosol (SOA), six sea-salt and marine organic bins  $(0.01-10 \ \mu m)$ , and five desert dust bins  $(0.2-48 \ \mu m)$ . Aerosol species are treated as externally mixed. Natural aerosol sources (desert dust, sea spray) are represented by online parameterizations, complemented by a biogenic emission scheme. Secondary organic aerosol formation and multiphase chemistry are explicitly treated, while anthropogenic emissions of primary particles and aerosol precursors are prescribed from contemporary inventories. Transport processes include advection, turbulent diffusion, sedimentation, and dry and wet deposition. Meteorological drivers, including wind fields, vertical diffusivities, and boundary-layer resistances, are directly provided by COSMO. In turn, the online coupling allows for feedback of modeled aerosol on radiation and clouds in COSMO. To better represent the microclimate of urban areas, the double-canyon urban canopy parameterization (DCEP) (Schubert et al., 2012) is included. This urban parameterization is based on the Building Effect Parameterization (BEP) by Martilli et al. (2002) and incorporates the radiative interactions between two neighboring urban canyons. The DCEP framework represents urban canopy structures through three categories of elements: ground, walls, and roofs. These elements are arranged within idealized double-street-canyon segments. Prior to the simulation, a preprocessing step derives the horizontal distribution of these segments in each model grid cell and determines probabilistic as well as geometric characteristics of the canopy elements, based on a high-resolution building geometry dataset available for Saxony. Within the model, DCEP calculates surface fluxes of momentum, heat, and turbulent kinetic energy (TKE), and additionally resolves the radiative transfer and thermal balance equations for the canopy components. Thereby, a realistic representation of the dynamics of heat transfer, radiation exchange, and airflow within urban street canyons is ensured.

# 2.1.1 Cloud Microphysics

In the standard setup, COSMO uses a simple single-moment bulk water continuity scheme to calculate the cloud microphysical 125 parameters (Doms et al., 2018). This scheme considers the specific mixing ratios of water in different phases and calculates cloud and precipitation formation based on these values. Cloud condensation nuclei (CCN) and ice nucleating particles (INP) concentrations are not considered. The scheme is less computationally demanding than other schemes, but likely does not fully account for certain small-scale microphysical processes. Here we use the two-moment bulk microphysics scheme developed 130 by Seifert and Beheng (2006b). It distinguishes between the five hydrometeor classes: cloud droplets, rain, ice crystals, snow, and graupel, and employs prognostic equations to estimate the mass densities and number concentrations of these hydrometeor particles. Therefore, it can provide more accurate predictions of cloud formation and precipitation than the simpler scheme (Seifert and Beheng, 2006a). In the standard setup of the two-moment scheme, the number of activated cloud droplets and ice particles is calculated using prescribed CCN and INP values, respectively. In this study, we replace these prescribed concentrations by calculating the activation of cloud droplets and ice particles from aerosol mass concentrations simulated by MUSCAT. 135 This dynamic approach allows the model to account for spatial and temporal variability in aerosol properties, which is expected to enable a more realistic representation of CCN and INP and should improve the simulation of aerosol-cloud interactions. We build upon the work of Weger et al. (2018), who included the effects of dust, soot, and organic carbon on ice particle formation and cloud droplet activation. Cloud droplet activation was computed using the parameterization for multiple aerosol types developed by Abdul-Razzak and Ghan (2000), which accounts for multiple soluble and insoluble aerosol species, repre-140 senting a multimodal aerosol size distribution. It uses the hygroscopicity parameter  $\kappa$ , which describes the relationship between particle hygroscopicity, dry diameter, and CCN activation (Petters and Kreidenweis, 2007). Experimentally determined values of  $\kappa$  range from greater than 1 for highly hygroscopic particles down to 0 for hydrophobic particles. In this study, the treatment of ice particle activation from Weger et al. (2018) is retained, while the cloud droplet activation 145 scheme is extended to include additional aerosol species, such as ammonium sulfate, ammonium nitrate, sulfate, organic carbon, elemental carbon, and two sea salt size classes. The corresponding  $\kappa$  values used in this study are listed in Table 1. Those species were chosen, as they are the most important natural and anthropogenic sources of cloud condensation nuclei (Pruppacher and Klett, 1997). For each species, the aerosol number size distribution is calculated from the simulated mass concentrations, using fixed values for radius and standard deviation, as in Genz et al. (2020). These size distributions are then used to calculate the number of activated particles using the Abdul-Razzak and Ghan (2000) parametrization, which links particle 150 size distribution and composition to the number of particles activated at a given supersaturation. The maximum supersaturation

# 2.1.2 Simulation Setup

updraft velocity.

Two cases with different meteorological characteristics, including flow patterns and boundary layer conditions, were selected.

Case I features a small-scale convective system approaching the region of interest with northerly winds in the evening of 13

is determined by the competition among aerosols for water vapor and depends on their composition, size distribution, and the

160

165

**Table 1.** Hygroscopicity parameters  $\kappa$  used in this study, based on (Genz et al., 2020). The values are based on findings from multiple laboratory experiments and modeling studies.

| Species                     | $\kappa$ |
|-----------------------------|----------|
| Dust (size classes 1-5)     | 0.14     |
| EC                          | 0        |
| Sulfate                     | 1        |
| Ammonium nitrate            | 0.54     |
| Ammonium sulfate            | 0.51     |
| POC + SOA                   | 0.14     |
| Sea salt (size classes 1-2) | 1.16     |

July 2019, while Case II shows a larger system with westerly inflow into the region in the evening of 20th July 2019. Despite these differences, both produced considerable precipitation across the region.

To simulate both cases we employ a 3-step one-way nesting strategy. The outer domain covers Europe with a grid resolution of 14 km (Fig. 1). There are 40 vertical layers with the highest layer at 22 km altitude. In this domain, COSMO-MUSCAT is run with the single-moment bulk water continuity scheme and without DCEP. The timestep is 80 s and the simulation starts 15 days prior to the actual event. This spin-up time is necessary to allow the model chemistry to develop. To keep the model meteorology close to the real synoptic situation, the meteorological fields are re-initialized for the next simulation cycle every 48 hours. The COSMO simulations are initialized and driven by ERA5 meteorological reanalysis data (Hersbach et al., 2020). The first MUSCAT cycle is initialized with an atmospheric chemical composition reanalysis product of CAMS (Copernicus Atmosphere Monitoring Service) (Inness et al., 2019), while subsequent cycles use aerosol and trace gas fields from the previous cycle to maintain temporal consistency in chemical composition. Natural primary aerosols, such as dust and sea salt, are computed within the MUSCAT model (e.g. Heinold et al. (2011)) using meteorological fields from COSMO, combined with surface property data. Emission of anthropogenic aerosols and trace gases are taken from the CAMS-REG inventory (Granier et al., 2019), while the temporal profiles of the main air pollutants and greenhouse gases are based on the CAMS-REG-TEMPO dataset (Guevara et al., 2021).

The next inner domain D1 covers Germany and is configured with a horizontal resolution of 2.3 km (Fig. 1). The vertical resolution includes 50 levels with the highest level at 22 km. As meteorological input, the COSMO-D2 reanalysis provided by the German Weather Service (DWD) is used. This high-resolution input is essential for resolving the small convective cells that are in the focus of this study. The simulation starts on 12 July 2019 at 00:00 UTC for Case I and at 19 July 2019 00:00 UTC for Case II, each simulation runs continuously for 48 hours without a restart and has a timestep of 20 s. The first 24 hours serve as a meteorological spin-up, while the subsequent 24 hours comprise the coupled chemistry-meteorology simulation. In this domain, the model is run with the two-moment bulk microphysics scheme. The chemical fields simulated form the coarser

**Figure 1.** (a) Model domains used in the simulation: Europe (14 km resolution), D1 (2.3 km), and D2 (1 km). (b) Inner model domain D2 showing geometric height of the earths surface above sea level. The cities Leipzig and Chemnitz are shown as filled areas in grey and light grey to indicate their urban extent.

European domain are interpolated onto the D1 grid as initial and lateral boundary conditions. Point source, detailed area, and traffic emissions that are used build on the GRETA dataset of the German Federal Environmental Agency (Umweltbundesamt, UBA) (Schneider et al., 2016).

The innermost domain D2 focuses on the greater metropolitan area of Leipzig, with a horizontal resolution of 1 km. The chemistry and meteorology for D2 are driven by the outputs from the D1 simulations, maintaining the same simulation time and vertical layers for both cases. The model in D2 operates with the two-moment bulk microphysics scheme, along with the urban parametrization scheme DCEP activated for the city of Leipzig. The simulation uses a timestep of 10 s, with output saved at 10-minute intervals. The domain is characterized by predominantly flat topography, bounded by the Erzgebirge along its southern edge. The highest peak of this mountain range rises to an altitude of about 1.2 km. The city of Leipzig lies at the center of the domain.

# 2.2 Sensitivity experiments

185

We conducted two different experiments for the Leipzig region. The first experiment (BASE) represents the reference simulation without any alterations. In the second experiment (NONURBAN), emissions within Leipzig are suppressed, whereas all other domain emissions are included. Comparing the two experiments helps us to isolate the effects of urban emissions on the cloud properties and specifically on precipitation. The chemical boundary and background concentrations were the same for both experiments. Surface parameters and DCEP parametrization stayed unchanged for Leipzig. For each experiment, we created an ensemble with five members, respectively, by varying the length of the meteorological spin-up run. The spin-up lengths

are 24h, 21h, 18h, 15h, 12h. This approach allows to assess the impact of slightly varying initial meteorological conditions on the results. By comparing multiple ensemble members, variability is captured more effectively, and consistent, non-stochastic patterns can be identified more clearly by statistical analysis. This increases the robustness of the findings in this study and helps to separate systematic signals from noise, providing greater confidence in the conclusions drawn from our simulations.

#### 200 2.3 Observational data

We compared our simulations with observed precipitation data from the RADKLIM dataset (Lengfeld et al., 2021), a radar-based precipitation climatology provided by the German Meteorological Service (Deutscher Wetterdienst, DWD). RADKLIM provides high-resolution data on a 1 - km spatial grid with a 5-minute temporal resolution covering entire Germany. The dataset is derived from 17 C-band Doppler radar systems and is offline-adjusted using daily gauge measurements from over 4,400 rain gauges that record both hourly and daily precipitation.

# 2.4 Analysis Method

205

# 2.4.1 Trajectory Analysis using LAGRANTO

To capture the movement of air masses influenced by urban emissions, we used the Lagrangian analysis tool LAGRANTO (Sprenger and Wernli, 2015). We calculated backward trajectories starting in the region of highest precipitation at the initial time for each ensemble member, using meteorological fields from the D2 output. For Case I, starting at 13 July 2019, 23:00 UTC, trajectories were traced backward for six hours. The horizontal start box covers approximately  $40 \times 40 \,\mathrm{km^2}$  with about 5 - km spacing between points. Vertically, the range extends from 1 to 4.5 km above ground level and is divided into 80 vertical levels. For Case II, starting at 20 July 2019, 22:00 UTC, the start points were similarly traced back for six hours. The horizontal box spans roughly  $40 \times 60 \,\mathrm{km}$  with 5 - km spacing, and vertically, it extends from 1 to 7.5 km subdivided into 100 levels. These spatial and vertical configurations ensure that the trajectories originate from sufficiently large and well-resolved volumes, capturing the key atmospheric dynamics of each precipitation event. To make sure that we only follow the Leipzig emissions, a filter region was defined as a three-dimensional box in longitude, latitude, and height. Only trajectories passing through this volume below 1 km altitude over the urban area of Leipzig were included in the analysis. By focusing on air parcels that interact with the urban environment prior to entering the precipitation area, this approach facilitates the attribution of aerosol–cloud interactions.

# 2.4.2 Moving Box Analysis

For analyzing meteorological conditions along the trajectories, we implemented a spatial averaging technique based on a calculated mean trajectory for each ensemble member. This method extracts vertical atmospheric profiles by averaging data within predefined rectangular boxes, each centered on a trajectory point. Each averaging box is defined in the COSMO rotated geographical coordinate system as a rectangular area with the trajectory point at its center and including the full vertical column of model layers. The boxes extend symmetrically in both horizontal directions (east—west and north—south). We selected dif-

**Figure 2.** Mean trajectories for Case I (a) and Case II (b). Pink–green shading shows the percentage difference in the mean of all aerosol species considered in the microphysics below 1000 m (Case I: 17:50–19:00; Case II: 18:20–20:00) between the NONURBAN and BASE experiments, covering the period from entering the urban area until ascent above 1000 m. Grey shading indicates the average trajectory height and the trajectories run in the direction towards higher altitudes. The filled areas mark the urban regions of Leipzig (dark grey) and Chemnitz (light grey). Grey boxes denote the rectangular averaging areas used to extract vertical profiles around each trajectory point.

ferent box sizes depending on the spatial scale of the respective precipitation system. For Case I, a box size of approximately  $15 \times 15 \,\mathrm{km^2}$  was chosen to fully enclose the smaller, localized precipitation structure (Fig. 2a). For Case II, a larger  $35 \times 35 \,\mathrm{km^2}$  box was used to capture the broader precipitation area associated with the more extensive system (Fig. 2b). This selection ensured that the dominant precipitation features were included in the spatial averaging for each case. At each trajectory point and time step, we averaged the number of activated cloud droplets, all hydrometeor species, and vertical wind components horizontally for each grid point within each box. This produced one vertical profile per ensemble member per time step. These profiles were then aggregated across all ensemble members to derive the ensemble mean, minimum, and maximum values.

# 2.4.3 Paired t-test Analysis

In order to evaluate the significance of differences between the ensemble simulations with and without urban emissions, we applied a paired t-test using the full ensemble. Each of the five ensemble members from the BASE simulation was paired with its counterpart from the NONURBAN simulation. The paired t-test assesses the null hypothesis that there is no systematic difference between the BASE and NONURBAN experiments. It was applied to fields that vary either in space and time (e.g., precipitation fields) or in time and height (e.g., vertical profiles along trajectories). The resulting p-values indicate whether the differences are statistically significant at the 90 % level (p 

**Figure 3.** Accumulated precipitation over a 4-hour period for both cases. Simulated precipitation ensemble mean from COSMO-MUSCAT (a, c) and radar-derived precipitation from RADKLIM (b, d) for Case I (top row) and Case II (bottom row).

### 3.2 Basic meteorological setting and precipitation amounts

# 3.2.1 Case I

In the first case, atmospheric conditions were dominated by a high-pressure system shifting from the Norwegian Sea toward the British Isles and a broad low-pressure system over Eastern Europe. This synoptic setup established a northerly flow across Germany, advecting cold, moist air masses into the region of interest. Several small-scale convective events developed over northern and eastern Germany. Northerly winds transported large amounts of sulfate into the model domain, reflecting long-range transport from coal combustion and industrial sources in Eastern Europe, and leading to elevated background concentrations between 2 and 6 km. The convective cell analyzed in this study formed east of Leipzig and propagated southwestward, passing over the city of Chemnitz. Low-level winds transported urban emissions from Leipzig toward the southeast, merging the cell at higher altitudes. The total accumulated precipitation associated with this system exceeded 9 mm over the course of a 4-hour period (Fig 3a). The convective cell began to intensify north of Chemnitz around 20:00 UTC and propagated southward at approximately 10 - 20 km h<sup>-1</sup> (Fig. 4). Maximum precipitation occurred between 21:00 and 21:30 UTC, when the precipitation core (approximately 5 km in diameter) reached intensities of 6–7 mm in 30 minutes, representing a typical heavy rainfall rate for convective summer systems. The core was surrounded by moderate precipitation of 2 - 4 mm (30 min)<sup>-1</sup>. The system then continued to propagate southward, gradually dissipating.

**Figure 4.** Ensemble mean of accumulated precipitation for case I every 30 minutes (left column) and ensemble mean differences (NONUR-BAN minus BASE) for four time steps (right column). Black contours indicate areas where differences are statistically significant (p

the precipitation differences remained consistently small at ±0.5 - 1 mm (30 min)<sup>-1</sup>. In the early stage of the system (Fig. 4b), a precipitation enhancement appeared on the western flank, coinciding with the intersection of the convective cell and the urban plume. This represents the first influence of urban emissions on the precipitation pattern. Simultaneously, the eastern flank exhibited reduced precipitation, indicating a localized atmospheric response to the urban air mass. In the following stage (Fig. 4d), the differences at both the western and eastern flanks persisted but shifted further south as the system propagated, demonstrating a continued evolution of the urban emission influence. In the next phase (Fig. 4f), the difference pattern shows a more mixed spatial pattern, suggesting complex interactions between the urban emissions and the convective system's internal dynamics. In the late stage (Fig 4h), the difference pattern shows a shift to more negative values, indicating that the urban emissions enhanced rainfall intensity at this later stage of the convective event. Additionally, the pattern, initially more positive and later more negative, suggests a southward shift of the precipitation core.

290 The influence of urban emissions on precipitation ranges between 10 - 20 %, comparing both experiments, although not all differences are statistically significant. Significant changes mainly occur at the edges of the system, where emissions can reach higher cloud layers, making their impact more pronounced in these regions. The precipitation core maintains its structure and intensity in both experiments, with variations only in the later stage of the convective system. This spatial distribution highlights the complex and heterogeneous response of convective systems to urban emissions.

# 3.2.2 Case II

305

In the second case, a shortwave trough moved northeastward over Benelux on the forward flank of a broad Atlantic upper-level low. Meanwhile, a cold front reached western Germany during the day. Upper-level flow from the southwest advected warm, moist subtropical air into the region, which enhanced both instability and vertical wind shear. The resulting overlap of warm, moist air aloft and cooler air from the west increased the potential for deep convection. These conditions favored the development of organized convective systems, including a squall line that propagated northeastward across northern and central Germany. The wind field promoted background concentrations dominated by secondary organic aerosols and ammonium sulfate.

The analyzed system exceeded 24 mm accumulated precipitation within a 4 h period, with a total horizontal extent of approximately 100 - 150 km and moved diagonal trough the upper model domain(Fig. 3c). The system reached its peak intensity between 20:00 and 20:30 UTC. The core of the system exhibited a multicellular structure with at least two pronounced local maxima (Fig. 5e). Rainfall rates within the core reached 15–20 mm (30 min)<sup>-1</sup>, classifying the event as a heavy precipitation event. Toward the edges of the system, rainfall intensities decreased to 2.5–10 mm (30 min)<sup>-1</sup>. Following its peak, the system continued to move northeastward and gradually dissipated.

Comparing the BASE and NONURBAN experiment reveals a temporal evolution of the urban influence of the system. Between 19:00 - 20:00 UTC, differences were relatively small below  $\pm$  1 mm (30 min)<sup>-1</sup> and randomly distributed (Fig. 5b & d). However, after 20:00 UTC when the system reaches its maximum intensity, a clear negative precipitation anomaly appeared in the NONURBAN experiment (Fig. 5f), indicating a precipitation enhancement in the presence of urban emissions. These differences concentrate mainly around one intensity core of the precipitation field, while other areas of the system are minimally

**Figure 5.** Ensemble mean of accumulated precipitation for case II every 30 minutes (left column) and ensemble mean differences (NONUR-BAN minus BASE) for four time steps (right column). Black contours indicate areas where differences are statistically significant (p

320

340

the same range (Fig. 5h)

These results demonstrate that urban emissions can enhance local precipitation intensity by 10 - 20 %. Mainly the precipitation core after 20:00 UTC is affected, showing a clear enhancement of precipitation by Leipzig's emissions. This demonstrates that urban modifications to cloud microphysics and convective processes require time to establish. Given the system's rapid propagation speed ( $\sim 50 - 70 \text{ km h}^{-1}$ ), its residence time over the Leipzig area remains limited ( $\sim 1 \text{ h}$ ). Nevertheless, this exposure proves sufficient for the system to obtain measurable urban modifications. These modifications continue to influence precipitation patterns as the system moves downstream, with enhanced precipitation effects observable 50 - 100 km downwind the urban source region.

# 3.3 Time series of hydrometeors and vertical structure

To determine how urban emissions may have influenced precipitation formation, we analyze air parcel trajectories that pass over Leipzig and extend into the precipitation region (Fig. 2). Along these trajectories, we examine the evolution of microphysical processes to assess how urban air masses were transformed and how these changes contributed to either enhanced or suppressed precipitation.

#### 3.3.1 Case I

330 In this Case, the mean trajectory originates approximately 30 km northwest of Leipzig and passes through the city diagonally between 17:50 and 18:40 UTC (Fig. 2a). It continues southeast for approximately 15 - 20 km following the low-level wind field. At this point, it encounters convective activity and is lifted from surface level to cloud height (2 - 4.5 km altitude), after which it moves straight south for another 80 km within the convective system. As the trajectory crosses the urban emission plume, which is characterized by up to 3 % enhanced aerosol concentrations relative to the NONURBAN experiment, the air mass picks up an urban signature. This influence is limited to the section of the trajectory prior to convective uplift.

Cloud droplets first appear along the trajectory at 18:30 UTC (Fig. 6a). The number of activated cloud droplets (QNC) increases steadily to a first maximum of 55 cm<sup>-3</sup> at 20:00 UTC, then continues rising to 65 cm<sup>-3</sup> after 21:00 UTC during the mature convective phase when strongest updrafts enhance supersaturation conditions. Subsequently, QNC declines to 10 cm<sup>-3</sup> as precipitation processes dominate particle removal. The NONURBAN experiment shows a slightly higher QNC peak, but differences remain within ensemble spread. The total amount of cloud condensate (QSUM) reaches a maximum of 260 mg m<sup>-3</sup> (Fig. 6b). Detailed analysis reveals that QSUM (see Supplements Fig. S1) is dominated by graupel, cloud and rain droplets. Indicating that the dynamics in the cloud lead to riming and rapid droplet growth. No significant differences exist between experiments for total condensate or individual hydrometeor categories. Precipitation begins after 19:00 UTC and increases until reaching a maximum precipitation rate of 0.6 mm (10 min)<sup>-1</sup> between 21:00 - 21:30 UTC (Fig. 6c). Both experiments achieve precipitation maxima simultaneously, the BASE experiment shows slightly higher rain rates after 21:30 UTC, though the differences remain within ensemble variability.

Overall, we found no substantial differences in the vertically averaged quantities between the experiments. All analyzed variables show nearly identical temporal evolution, with only minor differences that fell within the natural ensemble variability.

Figure 6. Box-averaged (15 x 15 km $^2$ ) and layer-thickness-weighted vertical averages (0 – 13 km) time series of (a) activated cloud droplet number concentration (QNC), (b) total cloud condensate (liquid water, ice, snow, graupel, rain, and hail; QSUM), (c) shows box-averaged rain rate. Lines indicate the ensemble mean, while shading represents the ensemble minimum–maximum range. Blue lines correspond to the BASE experiment, red to the NONURBAN experiment.

Therefore, we conducted a more detailed analysis of vertical variability to better understand the potential urban impact on convective processes that may not be captured by spatially and vertically averaged fields.

**Figure 7.** Box spatially averaged vertical profiles of number of activated cloud droplets QNC (a) and total cloud condensate QSUM (d), showing ensemble means for the BASE experiment for case I. (b) and (d) show the corresponding differences between the NONURBAN and BASE experiments for QNC and QSUM, respectively. Black crosses indicate levels where the differences are statistically significant at the 90 % confidence level.

The analysis of the number of activated cloud droplets reveals a distinct temporal evolution of cloud development. Cloud droplet activation initiates at 18:00 UTC and rapidly intensifies reaching concentrations of 150 - 250 cm<sup>-3</sup> with a vertical extent of 1 - 4 km (Fig. 7a). Between 20:00 - 22:00 UTC the system continuous to develop vertically, with QNC reaching its maximum vertical extent of up to 6 km. The hydrometeor distribution (Fig. 7c) shows a vertical extent of 6 km between 19:30 - 22:00 UTC. Peak QSUM exceeds 600 mg m<sup>-3</sup> at 3 km height at 21:30 UTC, corresponding with highest rain intensities. The later stage of the cloud is mainly dominated by ice phase processes, with substantial formation of graupel and snow.

Comparing the BASE and NONURBAN experiment reveals two distinct phases of urban influence on cloud microphysics. During the first phase between 18:30 - 20:30 UTC, the negative QNC difference values of 5-15 cm<sup>-3</sup> dominate (Fig. 7b), indicating that the urban aerosols act as additional condensation nuclei and the formation of cloud droplets is enhanced by 2-10 % compared to the pristine conditions. Alongside, QSUM shows a slight increase of 2-4 % during this initial phase.

In the second phase, after 20:30 UTC, the pattern in QNC reverses. Positive differences dominate across all vertical layers, showing a 2 - 7 % reduction in QNC compared to the NONURBAN experiment. However, this phase is characterized by statistically significant intensification of QSUM between 4 - 6 km altitude of 3 - 7 % (Fig. 7d). This is driven by enhanced ice microphysical processes including increased formation of graupel and snow within the cloud, while simultaneously reducing ice crystals and cloud droplets (see Fig. S3).

This two-phase pattern indicates a reorganization of the cloud microphysics caused by the urban emissions. The additional droplets formed in the first phase are lifted to higher altitudes where temperatures fall below the freezing level, which is located at approximately 2.8 km. At these altitudes, the increased droplet population enhances riming and aggregation processes, leading to more efficient formation of graupel and snow particles. This phase transition creates a positive feedback mechanism that intensifies the cloud dynamic.

**Figure 8.** Ensemble mean vertical profiles of positive (w pos) (a) and negative (w neg) (c) vertical velocities from the BASE experiment, spatially averaged over the analysis box for case I. Panels (b) and (d) display the deviations from the BASE case in the NONURBAN experiment. Statistically significant differences at the 90 % confidence level are marked with black crosses. Note: color pattern reversed for w neg, as values are negative.

The vertical redistribution of cloud condensate with increased concentrations above 4 km after 20 UTC and decreased concentrations below supports this interpretation. Urban emissions effectively enhance the ice phase development, promoting more efficient precipitation processes through intensified microphysical interactions. They also provide additional condensation nuclei in the initial phase, which subsequently undergo vertical transport and phase transformation, ultimately promoting more efficient precipitation formation through intensified ice microphysical processes.

To further support this conclusion, we analyzed the domain-averaged vertical profiles of the positive and negative vertical velocities (Fig.8). Here we see an enhancement of the updraft region in the same height as the increased number of hydrometeors. Supporting the theory that with urban emissions, the ice phase is enhanced, as more particles reach the freezing level, resulting in more freezing of droplets. Hence, more energy is released, which intensifies the updraft in the cloud, leading to more graupel and snow in the clouds. While Fig.8 (d) shows positive values of the downdraft region difference before 20:00 UTC, which temporally coincide with the enhancement of precipitation by urban emissions.

# 3.3.2 Case II

For Case II, the mean trajectory originated 40 - 60 km southwest of Leipzig and progresses northeastward, passing over the city between 18:00 - 19:00 UTC (Fig. 2b). North of Leipzig, eastward deflection for 30 minutes following the urban emissions plume before resuming a northeastward path. Between 20:00 and 21:00 UTC, convective lifting causes the trajectory to ascend vertically by approximately 2 to 4.5 km. Prior to convective uplift, the air mass along the trajectory acquires up to 2 % higher urban aerosol concentrations relative to the NONURBAN experiment.

The number of activated cloud droplets (QNC) increased between 19:00 - 20:00 UTC reaching a maximum of 45 cm<sup>-3</sup>, indicating strong CCN activation (Fig. 9a). The subsequent slower decay suggests coalescence processes becoming dominant

Figure 9. Box-averaged (35 x  $35 \text{ km}^2$ ) and layer-thickness-weighted vertical averages (0 – 13 km) time series of (a) activated cloud droplet number concentration (QNC), (b) total cloud condensate (liquid water, ice, snow, graupel, rain, and hail; QSUM), (c) shows box-averaged rain rate. Lines indicate the ensemble mean, while shading represents the ensemble minimum–maximum range. Blue lines correspond to the BASE experiment, red to the NONURBAN experiment.

over nucleation, consistent with the transition to precipitation development. Both experiments display similar temporal patterns, with minor differences occurring after the peak period (20:00 – 21:00 UTC). Although the BASE simulation shows slightly higher values, these remain within the ensemble spread. The total cloud condensate QSUM peaked at a maximum of

**Figure 10.** Box spatially averaged vertical profiles of number of activated cloud droplets QNC (a) and total cloud condensate QSUM (d), showing ensemble means for the BASE experiment for case II. (b) and (d) show the corresponding differences between the NONURBAN and BASE experiments for QNC and QSUM, respectively. Black crosses indicate levels where the differences are statistically significant at the 90 % confidence level.

1100 mg m<sup>-3</sup> at 20:00 UTC (Fig. 9b). Comparing the experiments, we see no significant difference. Consistent with Case I, QSUM is dominated by rain droplets and graupel. The onset of precipitation occurred around 19:00 UTC, coinciding with the increase in QNC and QSUM, intensifying to maximum rates of 2.7 mm (10 min)<sup>-1</sup> by 20:30 UTC (Fig. 9c). An observed lag of approximately 30 minutes between the peak in QNC and the maximum rain rate reflects the microphysical timescale necessary for cloud droplets to grow via condensation and collision-coalescence processes until precipitation-sized particles are formed.

Similarly to Case I, no pronounced influence of urban emissions was detected. The temporal evolution of all examined variables was largely consistent between experiments, with minor variations well within the ensemble spread. Here as well, we performed a more detailed examination of vertical variability to better assess potential subtle impacts that might not be evident in spatially and vertically averaged fields.

The BASE experiment shows a typical convective system evolution with a vertical extent reaching up to 13 km (Fig. 10a). The number of activated cloud droplets peaked around 20:00 UTC with maximum values exceeding  $150 \text{ cm}^{-3}$  at around 4 - 5 km height. The total hydrometeor concentration reached their maximum of  $1500 \text{ mg m}^{-3}$  at 20:00 UTC (Fig. 10c). During this peak intensity period, the system is characterized by dominant graupel production.

The experimental comparison between BASE and NONURBAN experiment reveals significant temporal variations in system response, with contrasting effects emerging across different phases of convective development. In the early phase of the system, a slight enhancement in hydrometeor number concentration is observed in the BASE experiment (Fig. 10d). This enhancement is primarily attributed to increased cloud droplet numbers above 6 km altitude and enhanced graupel formation shortly before the precipitation maximum. A slight increase in the QNC signal is also detected during this phase (Fig. 10b). At the time of peak precipitation (20:00 - 20:30 UTC), the BASE experiment shows significantly higher numbers of rain droplets, reflecting

**Figure 11.** Ensemble mean vertical profiles of positive (w pos) (a) and negative (w neg) (c) vertical velocities from the BASE experiment, spatially averaged over the analysis box for case II. Panels (b) and (d) display the deviations from the BASE case in the NONURBAN experiment. Statistically significant differences at the 90 % confidence level are marked with black crosses. Note: color pattern reversed for w neg, as values are negative.

enhanced rain processes during the system's most intense phase. After 20:00 UTC, the BASE experiment displays an increase of approximately 10% in activated cloud droplets. The difference pattern in QSUM shifts compared to the earlier phase and is characterized by increased graupel and snow formation of up to 5% (see Supplements Fig. S4). While the BASE experiment shows 8-12% more total ice content.

With urban emissions, cloud droplet activation is enhanced during the early phase (19:00 - 19:30 UTC) followed by rapid ice formation at higher altitudes. This efficient freezing process increases latent heat release, which intensifies graupel and snow formation, leading to intensified precipitation formation. The resulting enhanced precipitation leads to a stronger downdraft and suppresses the updraft essential for sustaining the convective system. Analysis of vertical velocity fields confirms significantly stronger downdraft regions by 5 - 10 % in the BASE experiment (Fig. 11d), supporting this mechanism. This negative feedback loop accelerates the system's decay, resulting in premature rain-out and earlier collapse of the convective cell. In contrast, the NONURBAN experiment is characterized by continued graupel and snow production in the later stage of the system (20:00 – 21:00 UTC). Weaker downdrafts allow sustained updraft activity, supporting ongoing aggregation and riming, leading to fewer cloud droplets and ice in the NONURBAN experiment. This is supported by increased updraft regions above 10 km and reduced downdraft regions after 20:30 UTC (Fig.11).

The comparison reveals different storm evolution patterns with significant implications for precipitation efficiency. When including the Leipzig emissions, the system is characterized by rapid intensification but has limited duration. Without urban emissions, the system has an extended lifetime through sustained up- and downdraft circulation which enables more complete conversion of cloud water to precipitation particles.

#### 4 Discussion and conclusion

This study examined the effects of urban emissions on two different convective systems passing by the city of Leipzig. We simulated two events with different synoptic settings using the coupled model system COSMO-DCEP-MUSCAT. For each event, ensemble simulations with five members were conducted for two emissions experiments. The BASE experiment serves as a control and is compared to the NONURBAN experiment, where the urban area of Leipzig has been assigned zero emissions, the rest being kept. Case I (13 July 2019) and Case II (22 July 2019) both occurred in the evening hours, with convective systems passing through the urban emission plume of Leipzig. Analyzed backward trajectories reveal that urban air masses were advected into the convective zone. The overall aerosol concentrations in both cases are dominated by background concentrations. However, the urban aerosol plume is visible in the ground levels below 1000 m.

In Case I, initial changes of precipitation were detected at the system edges, while in the later stages, an enhancement of icephase processes intensified the convective core, leading to enhanced rainfall. Overall, in this case, precipitation was modified
locally by 10-20%, but the structure of the convective system was not fundamentally altered. In Case II, urban emissions accelerated droplet activation and ice formation, which initially intensified precipitation by 10-20%. However, this led to stronger
downdrafts of 5-10%, which resulted in a premature decay of the system.

While both cases reveal effects in the ice phase, differences in thermodynamic settings, chemical background and convective intensities determine how these responses unfold. Case I, with moderate instability, allowed for a sustained enhancement of ice-phase processes and updrafts, whereas in Case II, stronger initial instability led to rapid development, with increased precipitation loading suppressing updrafts and weakening further ice processes. These findings highlight that urban aerosol effects depend on the prevailing atmospheric conditions and involve coupled microphysical–dynamical feedback that can either enhance or suppress precipitation. The feedback between microphysical processes and convective dynamics may either boost or counteract each other.

This study addresses several methodological recommendations proposed by Varble et al. (2023), such as an ensemble simulation to mitigate biases from single-run conclusions, prognostic aerosol representation to allow case-dependent evaluation of mixed-phase convection, and trajectory-based analysis for objective sampling of single convective clouds and aerosol-cloud interactions. Here, all these aspects and procedures are addressed to ensure robust and process-oriented results.

The five-member ensemble showed that internal variability is comparable in magnitude to urban emission-induced changes. In contrast, many previous studies have relied on single simulations, where apparent aerosol effects may partly reflect natural variability rather than microphysical responses to urban emissions. Only robust effects identified by t-tests are discussed, ensuring that reported modifications in precipitation and ice-phase processes are not artifacts of sampling variability. This highlights why, given the weak signals, ensembles are essential to accurately capture the small changes due to urban emissions.

The findings of this study are of similar magnitude to those reported in the multi-model study by Marinescu et al. (2021), who also examined CCN effects on convection, without specifically considering urban influences. Their study reproduced comparable updraft enhancement trends (5-15%), an indication for the latent heating mechanism. However, the COSMO version used in Marinescu et al. (2021) exhibited one of the weaker responses, likely reflecting limitations in its standard CCN treatment. In

contrast, the COSMO-MUSCAT system used here includes a coupled chemistry model that directly calculates cloud droplet activation from prognostic aerosol fields. This explicit aerosol-to-droplet activation process, combined with realistic spatial and temporal aerosol variability, enables a more detailed consideration of aerosol-cloud interactions.

A key advantage of the approach applied here is the use of backward trajectories, which allows us to isolate the effect of local emissions on cloud microphysics and precipitation. In comparison, grid-based or bulk analyses, where variables are averaged over large areas or volumes, often mask these process-level responses. Consequently, the trajectory framework complements the ensemble and prognostic aerosol approach by providing objective, process-based sampling of individual convective clouds and their interactions with urban aerosols.

However, limitations remain as only two convective events and five ensemble members per case were simulated, restricting generalizability. Future work should systematically explore the sensitivity to different emission scenarios and magnitudes, include additional convective events under varying synoptic conditions, and expand ensemble sizes to even more robustly quantify the effects of urban emissions across meteorological regimes. The results of this study underpin that urban emissions can modulate microphysical processes in convective systems and affect precipitation amounts.

Data availability. The COSMO-MUSCAT model output data are archived at DKRZ and are available upon request. The RADKLIM precipitation data is available online at https://opendata.dwd.de (last access: 21 August 2025).

Author contributions. FK added the aerosol-cloud feedback in COSMO-MUSCAT for anthropogenic aerosol. FK performed the COSMO-MUSCAT simulations, carried out the analysis, and wrote and designed the manuscript with contributions from all co-authors. BH provided support for the simulations and modeling advice. MQ and VM contributed to the project design and discussions. All co-authors reviewed the manuscript, with minor contributions to the text.

Competing interests. The authors declare that they have no conflict of interest.

Acknowledgements. The authors thank the COSMO-MUSCAT team for providing the model data and technical support for this project. We thank Axel Seifert (DWD) for his expert knowledge, advice and insights in 2M-Scheme/COSMO. We are grateful for computing time from the German Climate Computing Center (DKRZ). Recent German-wide emission data were provided by the German Environment Agency (Umweltbundesamt, UBA). Building geometries and orography (DGM1) are available from the State Enterprise for Geographic Information and Surveying Saxony (GeoSN). We thank the DWD for good cooperation and support. Minor grammar and style improvements were assisted by AI tools.

Financial support. This publication was funded by the Deutsche Forschungsgemeinschaft (DFG, German Research Foundation) under Germany's Excellence Strategy – EXC 2037 'CLICCS - Climate, Climatic Change, and Society' – Project Number: 390683824

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
