# Peer review of "Tracking the Impact of Urban Air Masses on Convective Precipitation: A Multi-Member Modeling Study"

_EGUsphere, 2025_

## Referee Comment (RC1)

Review of Tracking the Impact of Urban Air Masses on Convective Precipitation: A Multi-Member Modeling Study by Keil et al. (2025)

This study uses the coupled COSMO-DCEP-MUSCAT modeling system to investigate the impact of urban aerosols on deep convective clouds. Two case studies in the area of the German city of Leipzig are conducted. For each case, a five-member ensemble with different initial conditions is produced. To isolate the impact of urban aerosols, each ensemble is run with either all emissions or no urban emissions (but all other emissions). The authors find that the aerosol effects are not uniform between the two cases and the same pollution source can have different effects depending on the convective system.

Overall, I think the design of this study is strong and shows a clear vision by the authors on how to conduct a study trying to investigate aerosol impacts on deep convection. The approach using chemistry cloud coupling, an urban parameterization, an ensemble with multiple initial and boundary conditions, and tracking of the air mass are a strong foundation, which leads me to believe that this study can be a good addition to the literature and suitable for publication in ACP.

However, in my opinion there is a major flaw. In the introduction the authors correctly state "the influence of urban aerosols is more pronounced in less industrialized regions, where lower background aerosol concentrations amplify their impact on convection and precipitation processes". Then later they describe that "The [study] region is characterized by a mix of urban and industrial emission sources and an overall relatively high background aerosol concentration." In effect this leads to minimal differences between the simulations in terms of their aerosol loading. The differences are only around a few percent as shown by Figure 2. Many previous studies have used much more significant aerosol differences often with factors between 2 and 10.

Given these minimal differences in the aerosol loading, I am not fully convinced that the results the authors show are attributable to aerosol effects. Figures 4-6 and 9 suggest a spatial redistribution that mostly averages out. Only the vertical profiles seemingly show a more systematic aerosol effect. Furthermore, I do not really see much evidence for the statement "The same air pollution source can either delay, enhance or suppress convection [...]". Figures 6 and 9 show barely any difference between the aerosol setups, that would allow for such a statement. Please make this statement more accurate.

In my opinion, the authors can consider two things to address this concern. First, I suggest showing that there is no systematic difference in other convective parameters along the trajectory that might have influenced the convective evolution, i.e., make a figure similar to Figure 6 that looks at the evolution of CAPE for example. If there is no difference it would

increase my confidence that changes are attributable to aerosol indirect effects. Second, I suggest doing a similar analysis of convective systems that do not interact with the urban aerosols (or to a much lesser degree), to test whether these systems also change between the aerosol setups. You could produce Figures 4 and 5 (and potentially 7, 8, 10, 11) for these other systems. This could provide evidence that only the system that interacted with the urban aerosol showed changes.

If the authors can address this concern, I can suggest the manuscript for publication.

Below I list additional major and minor comments that the authors should address.

**Major comments**

- 1. I find the title to be somewhat misleading. Strictly, the study tracks the impact of urban aerosols on convective precipitation and not of urban air masses since the sensitivity experiments do not remove all urban influences. It might also make sense to add a geographical reference since aerosol impacts may vary substantially depending on location. Therefore, I suggest changing the title to "Tracking the Impact of Urban Aerosols on Convective Precipitation [over eastern Germany]: A Multi-Member Modeling Study". You can also consider other options.
- 2. Introduction: The description of the invigoration hypotheses (lines 32-53) is somewhat lacking. I suggest reworking these paragraphs by revisiting Varble et al. (2023) and the recent review by Fan et al. (2025) (and reference in both of those papers) for a balanced discussion of the state of the research on aerosol impacts on deep convection. Below, I mention a few points in particular.
  - The authors discuss aerosol cloud interactions in warm-phase clouds, in particular the first and second aerosol indirect effects. However, then they suddenly talk about the 'cold-phase invigoration hypothesis' without any transition, which might leave some readers confused.
  - A definition of invigoration is missing.
  - Cold-phase invigoration is described in such a way that it leaves the impression
    that it is scientific consensus. However, as the authors mention in the next
    paragraph the existence of a significant cold-phase invigoration effect is being
    debated. I would like to see more careful language when describing the potential
    mechanisms behind cold-phase invigoration.

- To my knowledge, the hypothesis is that a delay in precipitation development allows more condensate to be lofted to altitudes where it can freeze, not a delay in downdraft development.
- Warm-phase invigoration is mentioned without any reference or description. I
  think a more in-depth description of warm-phase invigoration is warranted,
  since anthropogenic emissions of ultra-fine aerosols may play a role in warmphase invigoration. Such aerosols are often from urban sources and thus it is
  possible that invigoration found in this study might be a result of warm-phase
  invigoration.
- 3. The manuscript includes some descriptions of results without showing evidence. Please consider including figures in the supplement such that the reader can verify the statements made by the authors. Since the authors make these descriptions, I suspect that these figures exist and it should not be much additional work to include them in the supplement and reference them in the text. Below, I list some examples, but the author should make sure to provide evidence for everything:
  - Lines 9-11, 446-447: The contrast in instability between the cases is only mentioned in the abstract and the final section but is not shown anywhere in the manuscript. Please show instability at least in the supplement.
  - Lines 155-158: Please show a figure of the meteorological characteristics of the two cases.
  - Section 2.4.1: A figure in the supplement could help to illustrate the trajectory analysis. You could use a figure showing some of the trajectories.
  - Lines 244-249: Figures might be needed to verify these statements about the convective development of the cases.
  - Lines 265-270, 296-301: Again, all these fields should be shown, such that the reader can verify the statements made here.
  - Line 421-422: I don't really see any evidence for an earlier decay of the system in the BASE experiment. Figure 9 basically shows no difference in the timing of the precipitation between the experiments. Please add a figure or better reference the figure that shows this result.

**Minor comments**

- 4. Lines 4, 59: It is probably better to say, "passing over" instead of only "passing".
- 5. Line 5: The sentence "[...], with five-member ensemble [...]" needs to be clarified, i.e. the ensemble needs to be described better. In its current form it reads more like five different emissions scenarios are compared and not five different IC/BC.

- 6. Line 20: Please add a reference for "[...] expected to further intensify precipitation events." or make the connection to the following sentence clearer.
- 7. Lines 27-28: The relevance of this sentence needs to be clearer. In which way do these types of aerosols relate to urban aerosols?
- 8. Lines 29 31: Please add references for this sentence.
- 9. Line 33: "narrower" needs to be clarified. In which way is the distribution narrower? I assume the authors mean more numerous and smaller cloud droplets.
- 10. Line 71: "[...] their findings [...]", please clarify whose findings are talked about.
- 11. Line 85: "two-moment bulk microphysics scheme"
- 12. Line 90: I would say Leipzig is a "city" not a "town".
- 13. Lines 105-106: Please add references to studies using this standard configuration.
- 14. Lines 125-129: The description of the single-moment scheme seems somewhat unnecessary and could be omitted. I think it would suffice to mention this in line 132: "[...] than the simpler *single-moment* scheme *employed in the standard COSMO setup* [...]" or something similar.
- 15. Lines 134-136: The authors could make it clearer that this capability was specifically developed for this study, if that is the case.
- 16. Line 137: Please rephrase to the following: "[...] a more realistic representation of CCN and INP with the goal of improving the simulation [...]" or something similar.
- 17. Line 151: Does supersaturation here refer to vapor supersaturation? Along those lines, the authors should describe how supersaturation is treated by the microphysics scheme. This is especially important because supersaturation plays a critical role in the warm-phase invigoration hypothesis and without predicting supersaturation in the microphysics scheme one cannot expect to accurately simulate warm-phase invigoration according to Fan et al. (2025). To my knowledge, the Seifert and Beheng scheme does use saturation adjustment, i.e., does not predict supersaturation.
- 18. Lines 155-158: In my opinion, more description about how these two cases were selected is needed. Currently, it reads like two random days with differing characteristics were selected without much explanation. Some questions I have: Why in particular were those two days selected? Were they chosen from a larger number of cases? Were they simulated better than other cases? Please add a few sentences on how these cases were selected.
- 19. Lines 174, 202: DWD is referred to as German Weather Service but later as the German Meteorological Service.
- 20. Section 2.4.1: You are using many more vertical levels for the trajectories than the model has. Does this actually provide any valuable additional information? Furthermore, do most trajectories pass over the city (see major comment 2, a figure would be helpful here)?

- 21. Figure 2: The moving boxes in the figure look quite rectangular and not square. Are they plotted correctly? Although it might be related to the map projection.
- 22. Line 232: Probably you mean "output time" or "every 10 minutes" instead of "every time step" which I would associate with the model time step (10 seconds).
- 23. Line 240: Please explain the selection of the significance level. I assume that it is because of the small sample size.
- 24. Line 272: "merging the cell at higher altitudes." I am not sure what the authors mean here.
- 25. Figures 4 and 5: The authors could mention in the caption that purple means more precipitation with urban aerosols. I might also suggest choosing a more colorblind-friendly color palette for the right column. Furthermore, the mean trajectories could be added to one of the panels.
- 26. Line 290: "The influence of urban emissions on precipitation ranges between 10 20 % [...]", this statement is vague, please clarify whether a decrease or increase is meant and whether total amounts or rates are referred to.
- 27. Line 306: The classification as a heavy precipitation event needs a reference.
- 28. Line 313: Please clarify around which precipitation core. Furthermore, "intensity core" seems like unusual language. I suggest changing to "precipitation core" and removing "of the precipitation field".
- 29. Line 334: "relative to the NONURBAN experiment", add figure reference.
- 30. Line 342: I would choose a different word than "significant", since no statistical testing was done in this specific case.
- 31. Line 358: I don't think I can agree with the statement that negative QNC differences dominate in the first phase. To me it does not really look like there is much dominance by either positive or negative differences. I would consider re-wording this statement.
- 32. Line 370: "[...] intensifies the cloud dynamic." This statement seems vague. Please clarify.
- 33. Lines 383-385: You used past and present tense in the first sentence. I am not able to understand what the authors are trying to say with the sentences afterwards.
- 34. Figures 7, 8, 9, 11: Please add "NONURBAN Base" above panels (b) and (d) similar to Figures 4 and 5.
- 35. Line 406: I suggest removing "dominant" or replacing it with another word such as "strong".
- 36. Line 407: I think "experimental" can be removed here.
- 37. Line 409: You probably mean Fig. 10b.
- 38. Line 436: "the rest being kept." This sentence is not finished.
- 39. Line 441: Please clarify what "intensified the convective core" exactly means.
- 40. Lines 452-460: This discussion is very valuable, thank you.

- 41. Lines 461-464: For the Marinescu et al. study, it is important to note that differences between aerosol number concentrations between the clean and polluted scenarios are 7-8 times. In this study the differences are only a few percent.
- 42. Lines 485-490: If possible, include links to these datasets.
- 43. Line 490: It might make sense to mention the specific AI tools.

**Typographical**

- 44. Line 175: "are the focus of this study."
- 45. Line 178: "simulated from the coarser"
- 46. Line 203: "covering the entirety of Germany"
- 47. Line 252: "a precise agreement"
- 48. Line 304: "through"
- 49. Line 316: Missing period.
- 50. Line 322: "downwind of the urban source region"
- 51. Line 330: "In this case"

**References**

Fan, J., Zhang, Y., Li, Z., Yan, H., Prabhakaran, T., Rosenfeld, D., & Khain, A. (2025). Unveiling aerosol impacts on deep convective clouds: Scientific concept, modeling, observational analysis, and future direction. Journal of Geophysical Research: Atmospheres, 130, e2024JD041931.

Varble, A. C., Igel, A. L., Morrison, H., Grabowski, W. W., & Lebo, Z. J. (2023). Opinion: A critical evaluation of the evidence for aerosol invigoration of deep convection. *Atmospheric Chemistry and Physics*, 23(21), 13791–13808. https://doi.org/10.5194/acp-23-13791-2023

---

## Author Comment (AC1)

**Responses to Reviewer #1**

**"Tracking the Impact of Urban Air Masses on Convective Precipitation: A Multi-Member Modeling Study"**

**by Keil et al.**

First of all, we thank the editor and reviewers for their thorough evaluation and constructive feedback on our manuscript. We have carefully addressed all comments and believe the revisions have substantially strengthened the paper. Below, we provide detailed point-by-point responses to each comment. Our responses are shown in blue text, and corresponding changes or additions to the revised manuscript are presented in *gray italic*.

**Reviewer #1**

This study uses the coupled COSMO-DCEP-MUSCAT modeling system to investigate the impact of urban aerosols on deep convective clouds. Two case studies in the area of the German city of Leipzig are conducted. For each case, a five-member ensemble with different initial conditions is produced. To isolate the impact of urban aerosols, each ensemble is run with either all emissions or no urban emissions (but all other emissions). The authors find that the aerosol effects are not uniform between the two cases and the same pollution source can have different effects depending on the convective system. Overall, I think the design of this study is strong and shows a clear vision by the authors on how to conduct a study trying to investigate aerosol impacts on deep convection. The approach using chemistry cloud coupling, an urban parameterization, an ensemble with multiple initial and boundary conditions, and tracking of the air mass are a strong foundation, which leads me to believe that this study can be a good addition to the literature and suitable for publication in ACP.

We thank the reviewer for the thorough evaluation of our manuscript and the positive assessment of our methodological approach. The reviewer's constructive feedback has helped us strengthen the manuscript, particularly regarding the clarification of our methodology, the interpretation of modest aerosol signals, and the discussion of case-specific responses. We have carefully addressed all comments and believe the revisions have substantially improved the clarity and scientific rigor of our work.

However, in my opinion there is a major flaw. In the introduction the authors correctly state "the influence of urban aerosols is more pronounced in less industrialized regions, where lower background aerosol concentrations amplify their impact on convection and precipitation processes". Then later they describe that "The [study] region is characterized by a mix of urban and industrial emission sources and an overall relatively high background aerosol concentration." In effect this leads to minimal differences between the simulations in terms of their aerosol loading. The differences are only around a few percent as shown by Figure 2. Many previous studies have used much more significant aerosol differences often with factors between 2 and 10.

The reviewer correctly identifies that our aerosol perturbations are modest compared to many previous studies. However, assessing whether such urban emission perturbations produce detectable convective responses was our central research question. While our introduction notes that urban effects are more pronounced in cleaner regions, the Leipzig case represents a

more challenging and realistic test case for a region with relatively high background aerosol concentrations. Whether realistic urban perturbations remain detectable above model noise in high-background aerosol environments was previously unknown and represents an important finding. Additionally, these realistic emission scenarios are particularly relevant for future cleaner cities, where urban aerosol perturbations are expected to be of comparable magnitude. The key methodological advance of this study is demonstrating that urban aerosol signals are detectable even under such small perturbation conditions when using our trajectory-based ensemble approach. A systematic investigation of emission scaling (including significantly larger perturbations) is the focus of our ongoing follow-up study, which we plan to submit at ACP in the next weeks. Preliminary results of the comprehensive sensitivity study (which is of such extent that its incorporation into this study is impractical) demonstrate a more profound impact of highly elevated emissions.

Given these minimal differences in the aerosol loading, I am not fully convinced that the results the authors show are attributable to aerosol effects. Figures 4-6 and 9 suggest a spatial redistribution that mostly averages out. Only the vertical profiles seemingly show a more systematic aerosol effect.

We are aware that the differences in accumulated precipitation and the time series are small. This is precisely why we conducted the vertical microphysical analysis. The vertical profiles reveal more pronounced changes in cloud structure and microphysics that are not immediately apparent in the time series. This vertical reorganization in cloud microphysics ultimately affects both the location and intensity of maximum precipitation

Furthermore, I do not really see much evidence for the statement "The same air pollution source can either delay, enhance or suppress convection [...]". Figures 6 and 9 show barely any difference between the aerosol setups, that would allow for such a statement. Please make this statement more accurate.

We agree with the reviewer. We have removed this statement from the abstract, as the observed differences do not support such a broad conclusion.

In my opinion, the authors can consider two things to address this concern. First, I suggest showing that there is no systematic difference in other convective parameters along the trajectory that might have influenced the convective evolution, i.e., make a figure similar to Figure 6 that looks at the evolution of CAPE for example. If there is no difference it would increase my confidence that changes are attributable to aerosol indirect effects.

We thank the reviewer for this suggestion. We have examined CAPE evolution along the trajectories and find no systematic differences in the ensemble means between BASE and NOURBAN scenarios (see Figure 1). This supports our interpretation that the observed hydrometeor changes are attributable to aerosol indirect effects rather than differences in the thermodynamic environment.

[Figure]

*Figure1: Box-averaged (case I: 15 x 15 km2; case II: 35 x 35 km2) and layer-thickness-weighted vertical averages (0 – 13 km) time series of CAPE for case I (a) and case II (b). Lines indicate the ensemble mean, while shading represents the ensemble minimum–maximum range. Blue lines correspond to the BASE experiment, red to the NONURBAN experiment.*

However, we note that CAPE itself can be modified by aerosol indirect effects through their influence on vertical wind and latent heat profiles. Therefore, while the similar CAPE evolution increases confidence in our results, we emphasize that our ensemble approach with 5 members and paired t-test analysis was specifically designed to account for internal variability. This statistical framework allows us to distinguish actual aerosol-induced signals (p < 0.1) from natural meteorological variability, thereby systematically addressing concerns about confounding factors in convective evolution. Also, we note that the urban parametrization remained unchanged between the base and nonurban experiments. Consequently, dynamic changes due to the urban parametrization can be ruled out.

Second, I suggest doing a similar analysis of convective systems that do not interact with the urban aerosols (or to a much lesser degree), to test whether these systems also change between the aerosol setups. You could produce Figures 4 and 5 (and potentially 7, 8, 10, 11) for these other systems. This could provide evidence that only the system that interacted with urban aerosol showed changes.

We agree that analyzing a convective system without urban aerosol influence would be valuable. To adress this concern we modified our trajectory filtering approach: instead of following the mean urban trajectory we followed the overall mean trajectory as seen in Figure 2. Hence, we can follow the convective system before it reaches urban influence.

[Figure]

*Figure 2: Trajectories for case I (a) and case II (b) showing every 5th trajectory in gray. Trajectories passing the urban area of Leipzig in red and mean trajectory in dark gray.*

Both cases show a clear temporal pattern in urban aerosol influence (case I Figure 3; case II Figure 4). In case I, clouds and updrafts developed before 18 UTC show no differences between base and nonurban simulations only after the system encounters the urban plume at approximately 18-20 UTC. In case II, the squall line exhibits cloud development before 19 UTC, with urban effects becoming apparent from 20 UTC onward. In both cases, the onset of urban influence coincides with the time when the convective systems intersect the urban aerosol plume, confirming that direct exposure to elevated urban aerosol concentrations is required for detectable modifications to convective processes.

We added these results to the appendix of the manuscript.

[Figure]

Figure 3: Top row is showing box spatially averaged vertical profiles of number of activated cloud droplets QNC (a) and total cloud condensate QSUM (c) along mean trajectory, showing ensemble means for the BASE experiment for case I. (b) and (d) show the corresponding differences between the NONURBAN and BASE experiments for QNC and QSUM, respectively. Bottom row is showing ensemble mean vertical profiles of positive (w pos) (a) and negative (w neg) (c) vertical velocities from the BASE experiment, spatially averaged over the analysis box along the mean trajectory for case I. Panels (b) and (d) display the deviations from the BASE case in the NONURBAN experiment. Black crosses indicate levels where the differences are statistically significant at the 90 % confidence level.

[Figure]

Figure 4: Top row is showing box spatially averaged vertical profiles of number of activated cloud droplets QNC (a) and total cloud condensate QSUM (c) along mean trajectory, showing ensemble means for the BASE experiment for case II. (b) and (d) show the corresponding differences between the NONURBAN and BASE experiments for QNC and QSUM, respectively. Bottom row is showing ensemble mean vertical profiles of positive (w pos) (a) and negative (w neg) (c) vertical velocities from the BASE experiment, spatially averaged over the analysis box along the mean trajectory for case II. Panels (b) and (d) display the deviations from the BASE case in the NONURBAN experiment. Black crosses indicate levels where the differences are statistically significant at the 90 % confidence level.

**Major comments**

1. I find the title to be somewhat misleading.

We agree with the review that the title was misleading. Therefore, we changed the title of the manuscript to: "Tracking Air Masses for Assessing the Effect of Urban Aerosols on Convective Precipitation: A Multi-Member Modeling Study "

2. Introduction: The description of the invigoration hypotheses (lines 32-53) is somewhat lacking. I suggest reworking these paragraphs by revisiting Varble et al. (2023) and the recent review by Fan et al. (2025) (and reference in both of those papers) for a balanced discussion of the state of the research on aerosol impacts on deep convection. Below, I mention a few points in particular.
   - The authors discuss aerosol cloud interactions in warm-phase clouds, in particular the first and second aerosol indirect effects. However, then they suddenly talk about the 'cold-phase invigoration hypothesis' without any transition, which might leave some readers confused.
   - A definition of invigoration is missing.
   - Cold-phase invigoration is described in such a way that it leaves the impression that it is scientific consensus. However, as the authors mention in the next paragraph the existence of a significant cold-phase invigoration effect is being debated. I would like to see more careful language when describing the potential mechanisms behind cold-phase invigoration.
   - To my knowledge, the hypothesis is that a delay in precipitation development allows more condensate to be lofted to altitudes where it can freeze, not a delay in downdraft development.
   - Warm-phase invigoration is mentioned without any reference or description. I think a more in-depth description of warm-phase invigoration is warranted, since anthropogenic emissions of ultra-fine aerosols may play a role in warm-phase invigoration. Such aerosols are often from urban sources and thus it is possible that invigoration found in this study might be a result of warm-phase invigoration.

We have substantially revised this section in the introduction to improve clarity and accuracy based on these suggestions:
*"In deep convective systems, warm-rain suppression is an integral component of aerosol invigoration mechanisms. Condensational invigoration occurs through enhanced condensation and latent heat release (Cotton and Walko, 2021). When large particles are abundant, they can get activated at the cloud base and suppress the warm rain formation, prolonging the condensation phase and releasing additional latent heat. In contrast, when ultrafine aerosol particles (UFPs), a significant component of urban emissions (Kumar, 2014), are abundant, the low concentration of large particles allows a rapid warm-rain formation, resulting in a high supersaturation, which then activates the UFPs, producing massive additional condensation and latent heat release (Fan et al., 2018). Freezing-induced invigoration is a more complex and uncertain mechanism where warm-rain suppression allows more liquid water to ascend to freezing levels, where latent heat release at high altitudes and ice processes (riming, deposition) can invigorate convection (Rosenfeld et al., 2008; Andreae et al., 2004). The net effect on convection depends on three competing processes. First, condensate loading during transport and depositional growth weakens updrafts (Fan and Khain, 2021); second, latent heat release from freezing at high altitudes*

*potentially strengthens updrafts; third, condensate offloading through hydrometeor sedimentation enhances buoyancy. "*

3. The manuscript includes some descriptions of results without showing evidence. Please consider including figures in the supplement such that the reader can verify the statements made by the authors. Since the authors make these descriptions, I suspect that these figures exist and it should not be much additional work to include them in the supplement and reference them in the text. Below, I list some examples, but the author should make sure to provide evidence for everything:

   - Lines 9-11, 446-447: The contrast in instability between the cases is only mentioned in the abstract and the final section but is not shown anywhere in the manuscript. Please show instability at least in the supplement.

The vertical difference in equivalent potential temperature between 850 and 500 hPa is used as a diagnostic measure of convective instability. Positive values indicate a decrease of $\theta_e$ with height and thus a potentially unstable stratification (Markowski and Richardson, 2011). As seen in Figure 5, case I exhibits near neutral stability at midday, suggesting that the convection in the afternoon was triggered by local instabilities. In contrast, case II shows extensive instability throughout southern Germany, which favored large-scale convective development.

[Figure]

*Figure 5: 12:00 UTC analysis of vertical differences in equivalent potential temperature between 850 and 500 hPa for case I (a) and case II (b).*

- Lines 155-158: Please show a figure of the meteorological characteristics of the two cases.

We show the meteorological characteristics of both cases in the following figure:

[Figure]

*Figure 6: Synoptic scale forcing for case I (a-c) and case II (d-f). Analysis at 12:00, 18:00 and 00:00 UTC showing 500 hPa geopotential height (gpdm; shading), sea level pressure (hPa, red contours), and 500 hPa wind barbs.*

- Section 2.4.1: A figure in the supplement could help to illustrate the trajectory analysis. You could use a figure showing some of the trajectories.

We added the following figure to the supplements showing the trajectory pathway and filtering for both cases:

[Figure]

*Figure 7: Trajectories for case I (a) and case II (b) showing every 5$^{th}$ trajectory in gray. Trajectories passing the urban area of Leipzig in light red and mean*

- Lines 244-249: Figures might be needed to verify these statements about the convective development of the cases.

As seen in Figure 8a case I is characterized by a more localized convective development, whereas the second case (Fig. 8b) involves a broader, more organized convective system.

[Figure]

Figure 8: RADKLIM accumulated precipitation over a 4-hour period for case I (a) and case II (b).

- Lines 265-270, 296-301: Again, all these fields should be shown, such that the reader can verify the statements made here.

The descriptions of the synoptic setups can be verified by Figure 6 and 9. Figure 6 shows 500 hPa geopotential height, sea level pressure, and 500 hPa wind barbs for both cases, illustrating the high-pressure system over the Norwegian Sea/British Isles and Eastern European low with northerly flow for case I, as well as the shortwave trough and southwesterly upper-level flow for case II. Figure 9 (850 hPa equivalent potential temperature, wind barbs, and sea level pressure) clearly shows the contrasting air mass advection patterns: cold air advection from the north in case I versus warm, moist subtropical air advection from the south in case II.

[Figure]

Figure 9: 12:00 UTC analysis of equivalent potential temperature at 850hPa, 850hPa wind barbs and sea level pressure (hPa, blue contours) for case I (a) and case II (b).

Information on aerosol background concentrations can be found in Figure 10, showing temporal (24h) and domain-averaged vertical profiles of the relevant chemical species.

[Figure]

*Figure 10: Spatially and temporally averaged vertical profiles of ammonium sulfate (light blue), sulfate (green), primary organic carbon (dark blue), and secondary organic aerosols (SOA; brown) for case I (a) and case II (b).*

- Line 421-422: I don't really see any evidence for an earlier decay of the system in the BASE experiment. Figure 9 basically shows no difference in the timing of the precipitation between the experiments. Please add a figure or better reference the figure that shows this result.

We thank the reviewer for this critical comment. Upon re-examination, we agree that our data does not clearly support the claim of "premature decay" or earlier system collapse. While Figure 11d in the manuscripts shows stronger downdrafts in the BASE experiment, Figure 9 in the manuscript demonstrates that precipitation timing is similar between experiments. We have revised the text to remove the claim of premature decay. The text was revised to:

Abstract: *"Under stronger initial instability, the urban emissions intensify the precipitation, leading to stronger downdrafts and weaker updrafts, altering the convective system's evolution compared to the zero urban emission scenario."*

Results: *"The resulting increased precipitation leads to a stronger downdraft and suppresses the updraft essential for sustaining the convective system. Analysis of vertical velocity fields confirms significantly stronger downdraft regions by 5 - 10 % in the BASE experiment (Fig. 11d), supporting this mechanism. In contrast, the NONURBAN experiment is characterized by continued graupel and snow production in the later stage of the system (20:00 – 21:00 UTC). Weaker downdrafts allow sustained updraft activity, supporting ongoing aggregation and riming, leading to fewer cloud droplets and ice in the NONURBAN experiment. This is supported by increased updraft regions above 10 km and reduced downdraft regions after 20:30 UTC (Fig.11). The comparison reveals different storm evolution patterns with significant implications for precipitation efficiency. When including the Leipzig emissions, the system produces enhanced*

*early precipitation accompanied by stronger downdrafts that suppress further updraft development. Without urban emissions, the system maintains more balanced updraft-downdraft circulation in the later stages, enabling sustained aggregation and riming processes. This sustained microphysical activity in the NONURBAN experiment could indicate a longer convective lifetime."*

Discussion: *"In case II, urban emissions accelerated droplet activation and ice formation, which initially intensified precipitation by 10-20%. However, this led to stronger downdrafts of 5-10%, which appeared to contribute to earlier decay of the system."*

**Minor comments**

4. Lines 4, 59: It is probably better to say, "passing over" instead of only "passing".

Corrected.

5. Line 5: The sentence "[…], with five-member ensemble […]" needs to be clarified, i.e. the ensemble needs to be described better. In its current form it reads more like five different emissions scenarios are compared and not five different IC/BC.

We revised the sentence:

*"Using the coupled COSMO-DCEP-MUSCAT modeling system, we simulate two convective events passing over the city of Leipzig, Germany, with experiments comparing total emissions to zero urban emissions, with five ensemble members for each setting."*

6. Line 20: Please add a reference for "[…] expected to further intensify precipitation events." or make the connection to the following sentence clearer.

Reference (Yan et al., 2024) added.

7. Lines 27-28: The relevance of this sentence needs to be clearer. In which way do these types of aerosols relate to urban aerosols?

We have revised the sentence.

*"Moreover, urban areas emit aerosols from traffic, heating, and industry that, depending on their size and chemical composition, serve as cloud condensation nuclei (CCN, particularly sulfates; Pruppacher and Klett, 1997) and ice-nucleating particles (INP, e.g. black carbon and organics; Burrows et al., 2022)."*

8. Lines 29 -31: Please add references for this sentence.

Added Rosenfeld (2008) as a reference.

9. Line 33: "narrower" needs to be clarified. In which way is the distribution narrower? I assume the authors mean more numerous and smaller cloud droplets.

We corrected this statement.

*"However, the resulting cloud droplet size spectrum tends to be more numerous and include smaller cloud droplets, which reduces coalescence and collision efficiency, thereby decreasing raindrop growth."*

10. Line 71: "[...] their findings [...]", please clarify whose findings are talked about.

Done.

*"Although focusing on a single event limits the generalizability of results to other meteorological contexts or regions, such process-level studies are essential for improving our understanding of aerosol-cloud interactions."*

11. Line 85: "two-moment bulk microphysics scheme"

Corrected.

12. Line 90: I would say Leipzig is a "city" not a "town".

Corrected.

13. Lines 105-106: Please add references to studies using this standard configuration.

Reference Wolke et al. 2012 added.

14. Lines 125-129: The description of the single-moment scheme seems somewhat unnecessary and could be omitted. I think it would suffice to mention this in line 132: "[...] than the simpler single-moment scheme employed in the standard COSMO setup [...]" or something similar.

We revised the description paragraph according to the reviewer's suggestions:

*"Here we use the two-moment bulk microphysics scheme developed by Seifert and Beheng (2006a, b). It distinguishes between the five hydrometeor classes: cloud droplets, rain, ice crystals, snow, and graupel, and employs prognostic equations to estimate the mass densities and number concentrations of these hydrometeor particles. Therefore, it can provide more accurate predictions of cloud formation and precipitation than the simpler single-moment scheme employed in the standard COSMO setup (Doms et al., 2018)."*

15. Lines 134-136: The authors could make it clearer that this capability was specifically developed for this study, if that is the case.

We have revised the sentence to formulate it more clearly:

*"We enhance the model configuration for this study by replacing these prescribed concentrations with calculated activation of cloud droplets and ice particles directly from aerosol mass concentrations simulated by MUSCAT."*

16. Line 137: Please rephrase to the following: "[...] a more realistic representation of CCN and INP with the goal of improving the simulation [...]" or something similar.

Done:

*"This dynamic approach allows the model to account for spatial and temporal variability in aerosol properties, which is expected to enable a more realistic representation of CCN and INP with the goal of improving the simulation of aerosol –cloud interactions."*

17. Line 151: Does supersaturation here refer to vapor supersaturation? Along those lines, the authors should describe how supersaturation is treated by the microphysics scheme. This is especially important because supersaturation plays a critical role in the warm-phase invigoration hypothesis and without predicting supersaturation in the microphysics scheme one cannot expect to accurately simulate warm-phase invigoration according to Fan et al. (2025). To my knowledge, the Seifert and Beheng scheme does use saturation adjustment, i.e., does not predict supersaturation.

Yes, supersaturation at line 151 refers to vapor supersaturation with respect to liquid water.

The reviewer correctly identifies saturation adjustment as a limitation (Lebo, 2012). However, we implement a modified Abdul-Razzak & Ghan (2000) activation scheme. The scheme explicitly calculates supersaturation during activation based on updraft velocity, aerosol size distribution, and composition, and enables in-cloud activation. Nevertheless, saturation adjustment is applied after each activation in the Seifert & Beheng (2006), which may dampen the feedback between qnc and condensation rates (Zhang, 2021). While our setup likely cannot fully capture condensational invigoration by ultrafine particles as described by Fan et al. (2018), other aerosol-cloud interaction mechanisms remain active, including effects on droplet size distributions, autoconversion, and ice processes through the explicit activation scheme. We have added clarification to the model description and expanded the discussion section to address this methodological limitation.

Section 2.1.1: *"In the standard setup of the two-moment scheme, the number of activated cloud droplets and ice particles is calculated using prescribed CCN and INP values, respectively, and saturation adjustment is applied. ....... To enable in-cloud conditions, already activated cloud droplets are treated as an additional aerosol mode with κ ≈ 0 and a diameter equal to the mean droplet size. This allows the activation scheme to distinguish between activated droplets and activatable aerosol at each model time step, enabling secondary nucleation in updrafts throughout the cloud depth."*

Discussion: *"Finally, we note that our microphysical setup, while including explicit aerosol activation and in-cloud nucleation, applies saturation adjustment. This approach may underestimate aerosol effects on convective intensity compared to fully explicit supersaturation schemes (Lebo et al., 2012; Zhang et al., 2021), though this dampening is likely modest given our realistic perturbations and focus on real mid-latitude cases rather than idealized deep convection. Future work could employ explicit supersaturation methods to provide upper-bound estimates and better constrain the range of urban aerosol effects on precipitation. Nevertheless, the results of this study underpin that modest urban emission variations can modulate microphysical processes in convective systems and affect precipitation amount."*

18. Lines 155-158: In my opinion, more description about how these two cases were selected is needed. Currently, it reads like two random days with differing characteristics were selected without much explanation. Some questions I have: Why in particular were those two days selected? Were they chosen from a larger number of cases? Were they simulated better than other cases? Please add a few sentences on how these cases were selected.

The two convective cases were systematically selected through a multi-step process: First, local newspaper archives were consulted to identify days with publicly reported heavy precipitation in the Leipzig region during summer 2019. These candidate events were then verified against the RADKLIM CatRaRE (Radar Climatology Catalog of Radar-based heavy rainfall Events) (Lengfeld et al., 2021) event catalog to confirm their meteorological significance. Finally, selected dates

were tested for availability and quality in the COSMO-D2 reanalysis dataset, which served as initial and boundary conditions for the simulations. The final selection of July13 and July 20, 2019 was made because these cases were documented both in public reports and CatRaRe database, were captured in the COSMO-D2 reanalysis and exhibited different synoptic patterns, allowing for testing of aerosol-cloud interactions under different meteorological regimes.

We added the following sentence on the selection method to the manuscript:

*"The cases were systematically selected through evaluation of public reports and the RADKLIM CatRaRe event catalog (Lengfeld et al., 2021) and whether the cases were captured by COSMO-D2 reanalysis."*

19. Lines 174, 202: DWD is referred to as German Weather Service but later as the German Meteorological Service.

Corrected.

20. Section 2.4.1: You are using many more vertical levels for the trajectories than the model has. Does this actually provide any valuable additional information? Furthermore, do most trajectories pass over the city (see major comment 2, a figure would be helpful here)?

The reviewer is correct that our initial trajectory vertical resolution exceeded the model's information content. We have revised our approach to use approximately 2× oversampling relative to model levels (20 trajectory levels for ~10 model levels in case 1; 50 for ~20 in case 2).

We have revised the manuscript accordingly:

*"The horizontal start box covers approximately 40 x 40 km$^2$ with about 5 km spacing between points. Vertically, the range extends from 1 to 4.5 km above ground level and is divided into 20 vertical levels. For case II, starting at 20 July 2019, 22:00 UTC, the start points were similarly traced back for six hours. The horizontal box spans roughly 40 x 60 km$^2$ with 5 km spacing, and vertically, it extends from 1 to 7.5 km subdivided into 50 levels."*

As seen in Figure 7, most backward trajectories from the precipitation regions bypass the city. That's why we filtered the trajectories to retain only those passing over Leipzig and ensure urban influence reaches the convection.

21. Figure 2: The moving boxes in the figure look quite rectangular and not square. Are they plotted correctly? Although it might be related to the map projection.

That's correct, the moving boxes appear rectangular rather than square in the figure, and this is indeed related to the map projection as suspected. The boxes are defined as squares in metric space. However, when plotted on a map with latitude-longitude axes (in degrees), they appear rectangular due to the convergence of meridians at mid-latitudes (~50°N). We added a kilometer scale to the figure to improve clarity.

22. Line 232: Probably you mean "output time" or "every 10 minutes" instead of "every time step" which I would associate with the model time step (10 seconds).

Corrected.

*"This produced one vertical profile per ensemble member every 10 minutes."*

23. Line 240: Please explain the selection of the significance level. I assume that it is because of the small sample size.

The choice of 90% confidence level is indeed motivated by our limited ensemble size. With only 5 ensemble members, statistical power is inherently limited. The 90% threshold (α=0.10) represents a pragmatic balance between detecting genuine aerosol effects and avoiding false positives.

24. Line 272: "merging the cell at higher altitudes." I am not sure what the authors mean here.

Changed to:

*"Low-level winds transported urban emissions from Leipzig toward the southeast, mixing into the convective cell at higher altitudes."*

25. Figures 4 and 5: The authors could mention in the caption that purple means more precipitation with urban aerosols. I might also suggest choosing a more colorblind- friendly color palette for the right column. Furthermore, the mean trajectories could be added to one of the panels.

We added the mean trajectories to the Figures and adapted the captions:

*"Purple shading indicates increased precipitation due to urban aerosols."*

We have verified the chosen color palette with a colorblind colleague who had no difficulties distinguishing the colors. However, we would appreciate the reviewer's suggestion for an alternative palette that might further improve accessibility.

26. Line 290: "The influence of urban emissions on precipitation ranges between 10 – 20 % [...]", this statement is vague, please clarify whether a decrease or increase is meant and whether total amounts or rates are referred to.

We have clarified the text:

*"The influence of urban emissions on local precipitation rates ranges between 10 - 20 %, with both increases and decreases depending on location, although not all differences are statistically significant."*

27. Line 306: The classification as a heavy precipitation event needs a reference.

We added DWD criteria as a reference: https://www.dwd.de/DE/wetter/warnungen_aktuell/kriterien/warnkriterien.html

*"Rainfall rates within the core reached 15 -20 mm per 30 minutes, classifying the event as heavy precipitation by German Weather Service (DWD) standards. "*

28. Line 313: Please clarify around which precipitation core. Furthermore, "intensity core" seems like unusual language. I suggest changing to "precipitation core" and removing "of the precipitation field".

Changed and clarification added:

*"These differences concentrate mainly around the central precipitation core, while other areas of the system are only marginally affected."*

29. Line 334: "relative to the NONURBAN experiment", add figure reference.

Reference Fig. 2a added.

30. Line 342: I would choose a different word than "significant", since no statistical testing was done in this specific case.

Changed to:

*"No substantial differences exist between experiments for total condensate or individual hydrometeor categories."*

31. Line 358: I don't think I can agree with the statement that negative QNC differences dominate in the first phase. To me it does not really look like there is much dominance by either positive or negative differences. I would consider re-wording this statement.

We agree that this statement was imprecise. Following the reviewer's comment on trajectory analysis (comment 11), we recalculated the trajectories, and the revised analysis shows a clearer and statistically significant negative QNC signal in the early phase. The updated figure now better supports this statement. We revised the sentence to:

*"During the first phase between 18:30 - 20:30 UTC, the negative QNC difference values of 5-10 cm$^{-3}$ are more prevalent (Fig.7b), indicating that the urban aerosols act as additional condensation nuclei and the formation of cloud droplets is enhanced by 2-10 % compared to the pristine conditions (see Fig. S3)."*

32. Line 370: "[...] intensifies the cloud dynamic." This statement seems vague. Please clarify.

We clarified the statement to:

*"This phase transition releases additional latent heat, which warms the surrounding air and strengthens the updrafts, creating a positive feedback loop between microphysics and dynamics."*

33. Lines 383-385: You used past and present tense in the first sentence. I am not able to understand what the authors are trying to say with the sentences afterwards.

We revised the two sentences to:

*"For case II, the mean trajectory originates 40 - 60 km southwest of Leipzig and progresses northeastward, passing over the city between 18:00 - 19:00 UTC (Fig. 2b). North of Leipzig, trajectories follow the urban emissions plume eastward for about 30 minutes before continuing northeastward."*

34. Figures 7, 8, 9, 11: Please add "NONURBAN – Base" above panels (b) and (d) similar to Figures 4 and 5.

We added these above panels to Figures 7, 8, 9, and 11 in the revised manuscript.

35. Line 406: I suggest removing "dominant" or replacing it with another word such as "strong".

Changed to:

*"During this peak intensity period, the system is characterized by strong graupel production."*

36. Line 407: I think "experimental" can be removed here.

We changed the sentence to:

*"The comparison between BASE and NONURBAN experiment reveals significant temporal variations in system response, with contrasting effects emerging across different phases of convective development."*

37. Line 409: You probably mean Fig. 10b.

Thank you for checking. We indeed refer to Fig. 10d, where the differences in hydrometeor mass mixing ratio are shown, rather than Fig. 10b which displays the differences in number of activated cloud droplets. However, the text before was misleading as it stated hydrometeor number concentration. We corrected this in the manuscript.

38. Line 436: "the rest being kept." This sentence is not finished.

We changed the sentence to:

*"The BASE experiment serves as a control and is compared to the NONURBAN experiment, where the urban area of Leipzig has been assigned zero emissions while all other emissions are kept unchanged."*

39. Line 441: Please clarify what "intensified the convective core" exactly means.

Clarified in the manuscript.

*"In case I, initial changes of precipitation were detected at the system edges, while in the later stages, an increase of ice-phase processes intensified the updraft, leading to enhanced rainfall."*

40. Lines 452-460: This discussion is very valuable, thank you.

We thank the reviewer for this positive feedback.

41. Lines 461-464: For the Marinescu et al. study, it is important to note that differences between aerosol number concentrations between the clean and polluted scenarios are 7-8 times. In this study the differences are only a few percent.

We revised this paragraph to:

*"The findings of this study generally align with those reported in the multi-model study by Marinescu et al. (2021), who also examined CCN effects on convection, without specifically considering urban influences. Their study reproduced comparable updraft enhancement trends (5 – 15 %), an indication for the latent heating mechanism, although they applied substantially larger CCN perturbations than the moderate urban emission changes examined here. The COSMO version used in Marinescu et al. (2021) exhibited one of the weaker responses compared to the other participating models, likely reflecting limitations in its standard CCN treatment. In contrast, the COSMO-MUSCAT system used here includes a coupled chemistry model that directly calculates cloud droplet activation from prognostic aerosol fields. This explicit aerosol-to-droplet activation process, combined with realistic spatial and temporal aerosol variability, enables a more detailed consideration of aerosol–cloud interactions."*

42. Lines 485-490: If possible, include links to these datasets.

We added the following information on data availability:

*"Recent German-wide emission data were provided on request by the German Environment Agency (Umweltbundesamt, UBA). Building geometries and orography (DGM1) are available from the State Enterprise for Geographic Information and Surveying Saxony (GeoSN; https://www.geodaten.sachsen.de/downloadbereich-digitale-3d-stadtmodelle-4875.html, last access 14 July 2020)."*

43. Line 490: It might make sense to mention the specific AI tools.

We added the used AI tools.

**Typographical**

44. Line 175: "are the focus of this study." Corrected.

45. Line 178: "simulated from the coarser" Corrected.

46. Line 203: "covering the entirety of Germany" Corrected.

47. Line 252: "a precise agreement" Corrected.

48. Line 304: "through" Corrected.

49. Line 316: Missing period. Corrected.

50. Line 322: "downwind of the urban source region" Corrected.

51. Line 330: "In this case" Corrected.

**References:**

- Abdul-Razzak, H. and Ghan, S. J.: A parameterization of aerosol activation: 2. Multiple aerosol types, Journal of Geophysical Research: Atmospheres, 105, 6837–6844, https://doi.org/10.1029/1999JD901161, 2000.
- Andreae, M. O., Rosenfeld, D., Artaxo, P., Costa, A. A., Frank, G. P., Longo, K. M., and Silva-Dias, M. A. F.: Smoking Rain Clouds over the Amazon, Science, 303, 1337–1342, https://doi.org/10.1126/science.1092779, 2004.
- Burrows, S. M., McCluskey, C. S., Cornwell, G., Steinke, I., Zhang, K., Zhao, B., Zawadowicz, M., Raman, A., Kulkarni, G., China, S., et al.: Ice-nucleating particles that impact clouds and climate: Observational and modeling research needs, Reviews of Geophysics, 60, e2021RG000 745, https://doi.org/10.1029/2021RG000745, 2022.
- Cotton, W. R. and Walko, R.: Examination of Aerosol-Induced Convective Invigoration Using Idealized Simulations, Journal of the Atmospheric Sciences, 78, 287 – 298, https://doi.org/10.1175/JAS-D-20-0023.1, 2021.
- Doms, G., Förstner, J., Heise, E., Herzog, H.-J., Mironov, D., Raschendorfer, M., Reinhardt, T., Ritter, B., Schrodin, R., Schulz, J.-P., and Vogel, G.: A Description of the Nonhydrostatic Regional COSMO Model. Part II: Physical Parameterizations, Deutscher Wetterdienst, https://doi.org/10.5676/DWD_pub/nwv/cosmo-doc_5.05_II, 2018.
- Fan, J. and Khain, A.: Comments on "Do Ultrafine Cloud Condensation Nuclei Invigorate Deep Convection?", Journal of the Atmospheric Sciences, 78, 329 – 339, https://doi.org/10.1175/JAS-D-20-0218.1, 2021.
- Fan, J., Rosenfeld, D., Zhang, Y., Giangrande, S. E., Li, Z., Machado, L. A. T., Martin, S. T., Yang, Y., Wang, J., Artaxo, P., Barbosa, H. M. J., Braga, R. C., Comstock, J. M., Feng, Z., Gao, W., Gomes, H. B., Mei, F., Pöhlker, C., Pöhlker, M. L., Pöschl, U., and de Souza, R. A.

F.: Substantial convection and precipitation enhancements by ultrafine aerosol particles, Science, 359, 411–418, https://doi.org/10.1126/science.aan8461, 2018.

- Kumar, P., Morawska, L., Birmili, W., Paasonen, P., Hu, M., Kulmala, M., Harrison, R. M., Norford, L., and Britter, R.: Ultrafine particles in cities, Environment International, 66, 1–10, https://doi.org/https://doi.org/10.1016/j.envint.2014.01.013, 2014.
- Lebo, Z. J., Morrison, H., and Seinfeld, J. H.: Are simulated aerosol-induced effects on deep convective clouds strongly dependent on saturation adjustment?, Atmospheric Chemistry and Physics, 12, 9941–9964, https://doi.org/10.5194/acp-12-9941-2012, 2012.
- Lengfeld, K., Walawender, E., Winterrath, T., and Becker, A.: CatRaRE: A Catalogue of radar-based heavy rainfall events in Germany derived from 20 years of data, Meteorol. Z., 30, 469–488, https://doi.org/10.1127/metz/2021/1088, 2021.
- Marinescu, P. J., Van Den Heever, S. C., Heikenfeld, M., Barrett, A. I., Barthlott, C., Hoose, C., Fan, J., Fridlind, A. M., Matsui, T., Miltenberger, A. K., et al.: Impacts of varying concentrations of cloud condensation nuclei on deep convective cloud updrafts—A multimodel assessment, Journal of the Atmospheric Sciences, 78, 1147–1172, https://doi.org/10.1175/JAS-D-20-0200.1, 2021.
- Markowski, P., & Richardson, Y. (2011). Mesoscale meteorology in midlatitudes. John Wiley & Sons.
- Pruppacher, H. R. and Klett, J. D.: Microphysics of Clouds and Precipitation, Atmospheric and Oceanographic Sciences Library, Springer, 2nd ed. edn., https://doi.org/10.1007/978-0-306-48100-0, 1997.
- Rosenfeld, D., Lohmann, U., Raga, G. B., O'Dowd, C. D., Kulmala, M., Fuzzi, S., Reissell, A., and Andreae, M. O.: Flood or drought: How do aerosols affect precipitation?, science, 321, 1309–1313, https://doi.org/10.1126/science.1160606, 2008.
- Seifert, A. and Beheng, K.: A two-moment cloud microphysics parameterization for mixed-phase clouds. Part 2: Maritime vs. continental deep convective storms, Meteorology and Atmospheric Physics, 92, 67–82, https://doi.org/10.1007/s00703-005-0113-3, 2006a.
- Seifert, A. and Beheng, K. D.: A two-moment cloud microphysics parameterization for mixed-phase clouds. Part 1: Model description, Meteorology and atmospheric physics, 92, 45–66, https://doi.org/10.1007/s00703-005-0112-4, 2006b.
- Wolke, R., Schröder, W., Schrödner, R., and Renner, E.: Influence of grid resolution and meteorological forcing on simulated European air quality: A sensitivity study with the modeling system COSMO–MUSCAT, Atmospheric Environment, 53, 110–130, https://doi.org/10.1016/j.atmosenv.2012.02.085, aQMEII: An International Initiative for the Evaluation of Regional-Scale Air Quality Models - Phase 1, 2012.
- Yan, H., Gao, Y., Wilby, R., Yu, D., Wright, N., Yin, J., Chen, X., Chen, J., and Guan, M.: Urbanization Further Intensifies Short-Duration Rainfall Extremes in a Warmer Climate, Geophysical Research Letters, 51, e2024GL108 565, https://doi.org/https://doi.org/10.1029/2024GL108565, e2024GL108565 2024GL108565, 2024.
- Zhang, Y., Fan, J., Li, Z., and Rosenfeld, D.: Impacts of cloud microphysics parameterizations on simulated aerosol–cloud interactions for deep convective clouds over Houston, Atmospheric Chemistry and Physics, 21, 2363–2381, https://doi.org/10.5194/acp-21-2363-2021,2021.

---

## Author Comment (AC2)

**Responses to Reviewer #2**

**"Tracking the Impact of Urban Air Masses on Convective Precipitation: A Multi-Member Modeling Study"**

**by Keil et al.**

First of all, we thank the editor and reviewers for their thorough evaluation and constructive feedback on our manuscript. We have carefully addressed all comments and believe the revisions have substantially strengthened the paper. Below, we provide detailed point-by-point responses to each comment. Our responses are shown in blue text, and corresponding changes or additions to the revised manuscript are presented in *gray italic*.

**Reviewer #2**

**General comment:**

The manuscript background and motivation to study urban aerosol effects on convective precipitation and the underlying microphysics is presented well. The modeling effort is well thought out and highly detailed including many aerosol sources, a chemistry model, urban land surface effects, an ensemble method, and so forth. I do find that 1km grid spacing for the inner domain to be on the borderline with regards to resolving the details of convective cells. While the methodology is reasonably sound, I am largely concerned that the "urban enhancement in aerosol concentration" is not really an enhancement since you only see a 2-3% increase. This is a very small change in aerosol for urban enhancement when you consider the other urban aerosol studies you cited in the introduction. Likewise, the changes you see in several of the figures are incredibly small. A 10-20% change of a very small number is still a very small number. As such, I am left wondering if the results are worth publishing. While the analysis seems sound, the aerosol change and the impacts are small. If the aerosol change was truly "urban-ish" and the changes were still small, then that would be worth sharing to the community. Finally, I think it is difficult to draw significant conclusions regarding warm or cold phase invigoration with such a small change in aerosol and very small change in W over a very small area. I think these conclusions are overstated given these issues. Please see more detail in the comments below.

We thank the reviewer for these thoughtful and critical comments. The reviewer is correct that the urban enhancement is small. However, this is a key finding of our study rather than a known limitation. Our motivation was to investigate urban aerosol effects under realistic Central European emission levels, without prior knowledge of the magnitude of these effects. Previous urban aerosol-convection studies have often examined idealized scenarios with large aerosol perturbations or compared highly polluted megacities to pristine backgrounds. In contrast, our goal is to investigate realistic urban emission perturbations from a European mid-sized city and determine whether such modest changes are sufficient to produce detectable effects on convective precipitation. This is exactly why we developed the moving box analysis and used an ensemble base statistical approach to systematically test for significant changes and rule out internal variability. The results show statistically significant changes. We believe this finding is valuable to the community precisely because it demonstrates that realistic urban aerosol changes can have detectable impacts on convection. Furthermore, we emphasize that our ensemble approach represents methodological advancement over previous studies. Most urban

aerosol-convection studies do not employ ensemble methods, making it difficult to distinguish aerosol signals from meteorological variability. Our 5-member ensemble with statistical significance testing provides a more robust framework for detecting subtle effects that might otherwise be masked by internal variability.

We have revised the manuscript to better communicate this motivation and the methodological strength of our approach.

We agree with the reviewer that our interpretation of "invigoration" may be too strong given the small magnitudes and spatial scales involved. We have revised the manuscript to adopt more cautious language.

**Specific comments:**

1.Model description: Does the microphysics scheme in COSMO used here (Seifert and Beheng 2006b) use a saturation adjustment scheme? If so, this is likely a problem for trying to assess aerosol impacts on cloud microphysics. In low aerosol conditions, supersaturation should not be fully consumed in each timestep and should be carried within the cloud.

The reviewer correctly notes that the Seifert and Beheng (2006) microphysics scheme employs a saturation adjustment, which is a known limitation for studies of aerosol–cloud microphysical interactions, as supersaturation is diagnostically removed after each advection time step.

In our simulations, cloud droplet activation is treated using the Abdul-Razzak and Ghan (2000) scheme, which explicitly predicts supersaturation during the activation step and determines cloud droplet number concentration as a function of updraft velocity, aerosol size distribution, and composition. Our implementation further allows *in-cloud activation* by representing pre-existing cloud droplets as a competing aerosol mode (with $\kappa \approx 0$), thereby enabling secondary nucleation in updrafts throughout the cloud depth, which is a key mechanism for warm-phase convective invigoration (Fan, 2018; Lebo, 2018).

Following activation, condensational growth is treated using saturation adjustment, consistent with Seifert and Beheng (2006). As shown by Lebo et al. (2012), this approach may damp aerosol effects on buoyancy in deep convection by enhancing condensation primarily at lower levels, leading to larger droplets and earlier precipitation formation. Zhang et al. (2021) quantified this effect for *major aerosol perturbations* (factor of 5–10 increases in CCN), finding reductions in aerosol-induced buoyancy responses by factors of approximately 2–3.

Several aspects are relevant for interpreting our results: (1) The in-cloud activation capability partially compensates for saturation adjustment by allowing secondary droplet formation in convective updrafts. (2) The aerosol perturbations considered here are substantially smaller than the factor of 5–10 CCN changes examined by Zhang et al. (2021), suggesting a weaker damping effect in our case. (3) Our study focuses on realistic mid-latitude convection, whereas the results of Lebo et al. (2012) are based on idealized simulations of deep continental convection, for which the quantitative applicability to specific real-case urban scenarios may be limited.

More generally, Seifert and Beheng (2006) argue that clouds usually relax rapidly toward thermodynamic equilibrium between water vapor and cloud droplets, making saturation adjustment a practical and robust approximation. Although conceptually paradoxical - since droplet activation depends on supersaturation that is subsequently eliminated by saturation

adjustment - the operator-splitting method allows supersaturation to control activation prior to condensation, providing an efficient and robust numerical treatment. At present, neither COSMO nor its successor ICON provides a fully explicit prognostic treatment of supersaturation suitable for general cloud microphysics.

Consistent with Grabowski and Morris (2017), differences between saturation-adjustment and explicit supersaturation treatments are expected to be small for shallow and moderately deep convection, where supersaturations typically remain below ~1% due to relatively weak updrafts.

Taken together, we interpret our results as a conservative (lower-bound) estimate of urban aerosol impacts on convective precipitation. We have clarified the activation and condensation treatment in the model description and expanded the discussion to explicitly acknowledge this limitation and its implications.

Section 2.1.1: *"In the standard setup of the two-moment scheme, the number of activated cloud droplets and ice particles is calculated using prescribed CCN and INP values, respectively, and saturation adjustment is applied. ....... To enable in-cloud conditions, already activated cloud droplets are treated as an additional aerosol mode with $\kappa \approx 0$ and a diameter equal to the mean droplet size. This allows the activation scheme to distinguish between activated droplets and activatable aerosol at each model time step, enabling secondary nucleation in updrafts throughout the cloud depth."*

Discussion: *"Finally, we note that our microphysical setup, while including explicit aerosol activation and in-cloud nucleation, applies saturation adjustment. This approach may underestimate aerosol effects on convective intensity compared to fully explicit supersaturation schemes (Lebo et al., 2012; Zhang et al., 2021), though this dampening is likely modest given our realistic perturbations and focus on real mid-latitude cases rather than idealized deep convection. Future work could employ explicit supersaturation methods to provide upper-bound estimates and better constrain the range of urban aerosol effects on precipitation. Nevertheless, the results of this study underpin that modest urban emission variations can modulate microphysical processes in convective systems and affect precipitation amounts."*

2.Line 196: How exactly do you vary the spinup lengths for the ensemble? Does this mean you vary the time of initialization or the period of analysis? It was a little unclear. I ask, because changing the initialization / model start time can sometime drastically alter how convective systems organize since the reanalysis data can have different degrees of truth at different times due to different amounts and quality of data input (soundings, surface stations, satellite obs, etc).

We vary the time of initialization. A standard run has 24 hours meteorological spin-up (only COSMO) and then 24 hours coupled simulation of COSMO-MUSCAT. To create the ensemble, we varied the 24h meteorological spin-up time und the 24h coupled simulation stayed unchanged. Importantly, all ensemble members use the same coarse domain (D1) simulation as input, ensuring that large-scale forcing remains consistent across the ensemble. We understand the reviewer's concern. However, with our approach all ensemble members simulate the same convective events but with slightly different initial atmospheric states due to the varying spin-up duration. The resulting spread captures internal atmospheric variability rather than uncertainties in reanalysis data quality. We clarified this in the revised manuscript.

*"For each experiment, we created an ensemble with five members, respectively, by varying the length of the meteorological spin-up run, while the 24h coupled COSMO-MUSCAT simulations remained unchanged. The spin-up lengths are 24h, 21h, 18h, 15h, 12h. All ensemble members*

*are initialized with the same D1 simulation, ensuring that the large-scale forcing remains consistent across the entire ensemble. This approach allows to assess the impact of slightly varying initial meteorological conditions on the results."*

3.Lines 201-205: A comparison to precipitation is mentioned here but no reference to a figure or an analysis of this. It would be good to mention how this data will be used.

Observational data are used to evaluate the model's general ability to reproduce the convective precipitation events. A more detailed comparison of simulated and observed precipitation can be found in Section 3.1. We added this information to section 2.3.

*"To evaluate the general model performance, we compared our simulations with observed precipitation data from the RADKLIM dataset (Winterrath et al., 2018), a radar-based precipitation climatology provided by the German Weather Service (DWD). RADKLIM provides high-resolution data on a 1 - km spatial grid with a 5-minute temporal resolution covering the entirety of Germany. The dataset is derived from 17 C-band Doppler radar systems and is offline-adjusted using daily gauge measurements from over 4,400 rain gauges that record both hourly and daily precipitation. A more detailed comparison of simulated and observed precipitation is presented in chapter 3.1."*

4.Figure 2: It is unclear what is being shown here. What is meant by "the mean of all aerosol species"? The mean of aerosol mass or number? Please clarify. And is the difference taken as Base – Nonurban? Finally +/- 3% seems like a rather small difference.

The mean of all aerosol species refers to the aerosol mass concentration averaged across all aerosol species considered in the two-moment microphysics scheme: five dust size classes, ammonium sulfate, ammonium nitrate, sulfate, organic carbon, elemental carbon, and two sea salt size classes. The plot shows the relative difference in aerosol mass between the NONURBAN and BASE scenarios, calculated as (NONURBAN - BASE) / BASE × 100%, temporally averaged over ~2 hours (case I: 17:50–19:00; case II: 18:20–20:00) and vertically averaged below 1000 m. The average specifically captures the period and altitude range where the trajectories pass through the urban aerosol plume before reaching the convective system. We have revised the figure caption to clarify these aspects.

*"Mean trajectories for case I (a) and case II (b). Pink‑green shading shows the percentage difference in aerosol mass concentration (NONURBAN - BASE) averaged across all aerosol species considered in the two-moment microphysics scheme (5 dust classes, ammonium sulfate, ammonium nitrate, sulfate, organic carbon, elemental carbon, 2 sea salt classes), vertically averaged below 1000 m and temporally averaged over the periods when trajectories pass through the urban aerosol plume (case I: 17:50‑19:00; case II: 18:20‑20:00). Grey shading indicates the average trajectory height and trajectories run in the direction towards higher altitudes. The filled areas mark the urban regions of Leipzig (dark grey) and Chemnitz (light grey). Grey boxes denote the rectangular averaging areas used to extract vertical profiles around each trajectory point."*

The ±3% enhancement represents the realistic urban aerosol signal for this region and case study. Our aim is to quantify the detectable urban effect under real-world conditions, rather than exploring idealized or artificially amplified perturbations.

5.Lines 227-228: Does this imply that Fig. 2a is supposed to be showing a localized precipitation structure? The figure caption indicates we are seeing differences in aerosols. Please clarify this text and the associated figure 2.

The references to Figure 2 refer to the trajectory paths, not the aerosol differences shown in the color shading. The figure provides geographical context for the trajectory analysis. We have revised the text to clarify this.

*"This selection ensured that the dominant precipitation features were included in the spatial averaging for each case. The associated backward trajectories are shown in Fig. 2a (case I) and Fig. 2b (case II)."*

6.Line 251: "differences in spatial and temporal resolution" between what things?

We revised the unclear formulation in the manuscript to:

*"Due to model simplifications, uncertainties in input and observational data, and limitations in parameterizing sub-grid-scale processes, a precise agreement between simulation and observation is not expected."*

7.Lines 255-259: Not sure I agree that the precipitation systems are well simulated; particularly for Case I in which not much precipitation was simulated compared to that observed. However, I understand model limitations and the difficulty in simulating case studies.

We acknowledge that case I shows quantitative deviations from observed precipitation, likely due to its smaller spatial scale which is more challenging to capture in convection-permitting simulations. However, both systems are qualitatively reproduced in terms of spatial extent, intensity range, and propagation (as stated in the text). Crucially, our analysis focuses on relative differences between emission scenarios rather than absolute agreement with observations.

8.Figure 3: What is happening regarding the wave-like structure to the precipitation from RADKLIM in Case II? Is this physical?

We appreciate this observation. Since this is not our area of expertise, we consulted radar experts at the Meteorological Institute at the University of Hamburg, who confirmed that the wave-like structure visible in the RADKLIM observations for case II is a radar artifact rather than a physical precipitation feature. This artifact occurs when convective systems move rapidly relative to the radar scan. Between consecutive radar scans, the fast-moving system shifts position significantly, creating an apparent wave-like pattern in the composite product. This is a known limitation of radar-derived precipitation products when observing rapidly propagating convective systems.

9.Line 334: A 3% enhancement in aerosol concentration hardly seems like an urban influence. At least some of the urban aerosol studies cited in the introduction showed that urban aerosol enhancements can increase aerosol number concentrations by an order of magnitude. So 3% seems quite small. Your comments on this would be helpful.

The 3% enhancement reflects the realistic urban aerosol perturbation from a mid-sized Central European city (Leipzig, ~600,000 population) embedded in high regional background aerosol levels. Unlike previous studies of megacity impacts with order-of-magnitude increases, our focus is on detecting whether such modest but realistic urban signals are distinguishable from meteorological variability and model noise in a region not previously examined for urban aerosol-precipitation effects. This required developing a refined trajectory-based analysis

methodology to isolate the urban signal. The modest aerosol enhancement, which was not known prior to this analysis and represents a novel finding for Central European settings, motivated a follow-up study (currently in preparation) systematically varying emission strengths to quantify precipitation sensitivity across a wider range of aerosol perturbations. We revised the manuscript to more clearly communicate our motivation.

10. Figure 6: Both simulations seem very clean with very low cloud droplet concentrations, and there's almost no difference in the droplet number. I would be hard pressed to call this an urban enhancement compared to the studies you cited earlier. Given on a 3% change in aerosol concentration enhancement, this is perhaps expected.

We thank the reviewer for this comment, which highlights an important aspect of our analysis methodology. The clean appearance results from three layers of averaging: (1) ensemble averaging over 5 members, (2) spatial averaging within the trajectory boxes, and (3) layer-thickness-weighted vertical averaging (0 – 13 km), also the inherent spread in cloud positions across ensemble members, where clouds do not occur at exactly the same locations despite similar synoptic conditions. This averaging dilutes the local peak values.

To better illustrate the actual range of QNC values, we have created Figure 1 showing the frequency distributions of QNC within the trajectory boxes. These reveal that the most frequent values reach ~300 cm$^{-3}$ with maxima up to 700 cm$^{-3}$ for case I and up to 1400 cm$^{-3}$ for case II. These values are substantial and consistent with polluted continental conditions.

[Figure]

*Figure1: Frequency distribution of QNC along trajectory boxes for case I (a) and case II (b).*

While ensemble-mean QNC changes are small in Fig. 6, Fig. 7 in the manuscript shows that urban aerosols cause a vertical redistribution of hydrometeors within the cloud rather than uniform changes. This reorganization affects precipitation formation, as seen in the statistically significant changes in rain and graupel. The ~3% aerosol enhancement is sufficient to trigger these microphysical redistributions, which we believe is worth reporting.

11. Line 388: Could you please include a plot or two of the representative aerosol concentrations. Given your maximum droplet number of 45/cm3 (which is quite low and quite clean for a continental case) it would be good to know what fraction of aerosols are activating.

As discussed in our response to comment #10, the maximum droplet number of 45 cm$^{-3}$ shown in Fig. 6 results from triple averaging (ensemble, spatial, and vertical) by which many zeros enter the averaging procedure. The actual QNC values are substantially higher, with most frequent values of ~300 cm$^{-3}$ and maxima up to 600 cm$^{-3}$ (case I) and 1300 cm$^{-3}$ (case II) as shown in the frequency distribution figure. These values are representative of polluted continental conditions.

12. Line 403: It is unclear how Figure 10a indicates a convective system with a vertical extent reaching up to 13km. The feature in this figure panel tops out at 8-9km.

Figure 10a in the manuscript shows the number of activated cloud droplets (liquid phase), which indeed reach up to ~8-9 km. The full vertical extent of the convective system (up to 13 km) is evident when including the ice phase, as shown in Figure 10c in the manuscript. We have corrected the figure reference in the revised manuscript.

13. Line 405: It would be better to refer to the value of 1500 mg/m3 as a mass mixing ratio instead of a "total hydrometeor concentration". Further, mass mixing ratios are typically reported in units of g/kg.

We have revised the terminology to 'mass mixing ratio' and converted all hydrometeor values to mass mixing ratios with unit [g/kg]. All figures and text have been updated in the revised manuscript.

14. Line 412: What figure shows the higher number of rain drops? Also please use "rain drops" instead of "rain droplets".

We have corrected the terminology to 'rain drops' and added a reference to the supplementary material showing the rain drop number concentrations.

*"At the time of peak precipitation (20:00 - 20:30 UTC), the BASE experiment shows significantly higher numbers of rain drops, reflecting enhanced rain processes during the most intense phase of the system (see Fig. S4)."*

15. Line 414: A 10% increase in droplet number seems very small and is well below the change seen in most urban aerosol and convection studies. I still think it's overstated to call this an urban enhancement, when very substantial urban enhancements are noted in the literature.

We agree with this assessment. The observed changes are modest compared to megacity studies, which reflects our focus on realistic perturbations from a mid-sized city. We have revised the text to use more conservative terminology (e.g., 'increase' rather than 'enhancement') to avoid overstating the magnitude of the urban signal relative to the established literature.

16. Lines 460-462: Another key difference from the multi-model simulations in Marinescu et al. (2021) is that the aerosol loading from Clean to Polluted in that study was close to an order of magnitude difference. Here your differences are only 2-3%. I find it difficult to say these are comparable.

We have revised the text to appropriately reflect the substantial differences in aerosol perturbation magnitudes between the studies.

*"The findings of this study generally align with those reported in the multi-model study by Marinescu et al. (2021), who also examined CCN effects on convection, without specifically considering urban influences. Their study reproduced comparable updraft enhancement trends (5 – 15 %), an indication for the latent heating mechanism, although they applied substantially larger CCN perturbations than the moderate urban emission changes examined here. The COSMO version used in Marinescu et al. (2021) exhibited one of the weaker responses compared to the other participating models, likely reflecting limitations in its standard CCN treatment. In contrast, the COSMO-MUSCAT system used here includes a coupled chemistry*

*model that directly calculates cloud droplet activation from prognostic aerosol fields. This explicit aerosol-to-droplet activation process, combined with realistic spatial and temporal aerosol variability, enables a more detailed consideration of aerosol–cloud interactions."*

**References:**

- Abdul-Razzak, H. and Ghan, S. J.: A parameterization of aerosol activation: 2. Multiple aerosol types, Journal of Geophysical Research: Atmospheres, 105, 6837–6844, https://doi.org/10.1029/1999JD901161, 2000.
- Fan, J., Rosenfeld, D., Zhang, Y., Giangrande, S. E., Li, Z., Machado, L. A. T., Martin, S. T., Yang, Y., Wang, J., Artaxo, P., Barbosa, H. M. J., Braga, R. C., Comstock, J. M., Feng, Z., Gao, W., Gomes, H. B., Mei, F., Pöhlker, C., Pöhlker, M. L., Pöschl, U., and de Souza, R. A. F.: Substantial convection and precipitation enhancements by ultrafine aerosol particles, Science, 359, 411–418, https://doi.org/10.1126/science.aan8461, 2018.
- Grabowski, W. W., and H. Morrison, 2017: Modeling condensation in deep convection. J. Atmos. Sci., 74 (7), 2247–2267, doi:10.1175/JAS-D-16-0255.1.
- Lebo, Z. J., Morrison, H., and Seinfeld, J. H.: Are simulated aerosol-induced effects on deep convective clouds strongly dependent on saturation adjustment?, Atmospheric Chemistry and Physics, 12, 9941–9964, https://doi.org/10.5194/acp-12-9941-2012, 2012
- Lebo, Z., 2018: A Numerical Investigation of the Potential Effects of Aerosol-Induced Warming and Updraft Width and Slope on Updraft Intensity in Deep Convective Clouds. J. Atmos. Sci., 75, 535–554, https://doi.org/10.1175/JAS-D-16-0368.1.
- Marinescu, P. J., Van Den Heever, S. C., Heikenfeld, M., Barrett, A. I., Barthlott, C., Hoose, C., Fan, J., Fridlind, A. M., Matsui, T., Miltenberger, A. K., et al.: Impacts of varying concentrations of cloud condensation nuclei on deep convective cloud updrafts—A multimodel assessment, Journal of the Atmospheric Sciences, 78, 1147–1172, https://doi.org/10.1175/JAS-D-20-0200.1, 2021.
- Seifert, A. and Beheng, K.: A two-moment cloud microphysics parameterization for mixed-phase clouds. Part 2: Maritime vs. continental deep convective storms, Meteorology and Atmospheric Physics, 92, 67–82, https://doi.org/10.1007/s00703-005-0113-3, 2006a.
- Winterrath, T., Brendel, C., Hafer, M., Junghänel, T., Klameth, A., Lengfeld, K., Walawender, E., Weigl, E., and Becker, A.: Radarklimatologie aus quasi-angeeichten 5-Minuten-Niederschlagsraten Version 2017.002 Gerasterte Niederschlagswerte für Deutschland version v2017.02, https://doi.org/10.5676/DWD/RADKLIM_YW_V2017.002, 2018.
- Zhang, Y., Fan, J., Li, Z., and Rosenfeld, D.: Impacts of cloud microphysics parameterizations on simulated aerosol–cloud interactions for deep convective clouds over Houston, Atmospheric Chemistry and Physics, 21, 2363–2381, https://doi.org/10.5194/acp-21-2363-2021,2021.